# Spontaneous droplet transport on shape-evolving microfiber rails

Shiyu Wang [1], Ying Zhou [1], Wenchang Zhao [1], Yanhong Li [1], Shuxian Tang [1], Ting Si [2] ✉ & Pingan Zhu [1,3] ✉

Directional droplet motion on solid surfaces, governed by Gibbs' theory of minimizing surface free energy, traditionally relies on static surface gradients. However, such approaches face intrinsic constraints, including fixed transport directions and interdependence between transport distance and speed. Here, we introduce shape-evolving microfiber rails (SEMRs) that enable spontaneous droplet transport on initially non-gradient surfaces. Inspired by a domino-like mechanism, the SEMRs dynamically generate cascading gradients through shape deformations in response to droplet-surface interactions, diverging from conventional static-gradient designs. This adaptive strategy allows for steerable droplet motion, independent control over transport distance and velocity, and design flexibility for on-demand performance optimization. The SEMR's versatility heralds advanced applications across analytical chemistry, cargo transport, electronics, and diagnostics, paving the way for advanced intelligent fluidic systems.

Directional liquid transport is a fundamental phenomenon with broad applications, including microfluidics, chemical reactions, fog harvesting, liquid-liquid separation, and enhanced heat transfer[1–15]. Nature provides inspiring examples of such transport on surfaces such as spider silk[16], desert beetles[17–19], cactus spines[18,20], pitcher plants[21–24], *Araucaria* leaves[1], and *Crassula muscosa*[25].

Directional liquid transport manifests in two primary forms: directional liquid spreading and directional droplet transport. The former involves the guided movement of a liquid film along wettable surfaces, requiring continuous liquid feeding to sustain the advancing meniscus. In contrast, directional droplet transport entails the motion of discrete droplets on solid surfaces, free from the constraints of surface wettability and continuous liquid supply. This droplet-based transport underpins critical technologies such as digital microfluidics, micro-reactors, droplet carriers, and liquid robots[26–31].

The theoretical basis for droplet motion is rooted in Gibbs' foundational work on wetting behavior and surface tension[32], which dictates that droplets move on solid surfaces to reduce surface free energy. Building on this principle, conventional strategies for achieving directional droplet transport have relied on surfaces engineered

with intrinsic gradients, which encompass variations in surface chemistry[33–36], roughness[5,37], stiffness[38], and geometry[35,39–41]. These gradients create a net driving force that propels droplets along the surface in a pre-defined direction. Despite significant progress, static-gradient surfaces remain fundamentally constrained. Because their gradients are permanently fixed once fabricated, they can typically improve only one or a limited subset of performance metrics—such as directionality, velocity, or distance—often at the expense of others[42–45]. Consequently, one must compromise between competing design goals, as direction, speed, distance, and volume of droplets cannot be independently or dynamically controlled. Overcoming these constraints requires strategies that transcend the inherent limitations of static-gradient designs. Yet, a key question remains unresolved: can directional droplet transport be realized on surfaces devoid of intrinsic gradients?

We introduce shape-evolving microfiber rails (SEMRs), an initially non-gradient system designed to enable spontaneous droplet transport. Unlike conventional static-gradient surfaces, SEMRs self-generate and continuously evolve geometric gradients through droplet-surface interactions, establishing a feedback-driven mechanism for motion.

[1]Department of Mechanical Engineering, City University of Hong Kong, Hong Kong, China. [2]Department of Modern Mechanics, University of Science and Technology of China, Hefei, China. [3]Shenzhen Research Institute of City University of Hong Kong, Shenzhen, China. ✉e-mail: tsi@ustc.edu.cn; pingazhu@cityu.edu.hk

This self-adaptive behavior allows for on-demand tuning of all key transport parameters—direction, velocity, distance, and volume of droplets—within a single system. By integrating multiple performance controls into a unified and reconfigurable surface, SEMRs overcome the inherent limitations of static designs. With their dynamic adaptability and broad functionality, SEMRs offer a versatile foundation for next-generation microreactors, fluidic logic circuits, diagnostics, and autonomous liquid handling systems. This study thus establishes a paradigm of self-evolving liquid transport surfaces, bridging the gap between passive gradient engineering and active fluidic control.

## Results

### Design principle

To overcome the limitations imposed by static surface gradients in directional droplet transport, we draw inspiration from the domino effect to design the SEMR system, composed of two parallel microfibers (Fig. 1a). Analogous to a line of dominos, where each falling domino triggers the next, the initially non-gradient SEMR dynamically generates geometric gradients upon contact with a water droplet, propelling it in a steerable direction. As the droplet moves, it interacts with adjacent microfiber segments, continuously evolving gradients through surface deformation and maintaining its motion (Fig. 1a). This design eliminates pre-existing gradients or external energy input, enabling the SEMR to adaptively respond to the droplet. Therefore, it integrates the advantages of both passive and active strategies for directional droplet transport.

The design of SEMRs necessitates microfibers that exhibit (i) responsiveness to the transporting droplets (Fig. 1b), and (ii) structural asymmetry at the macro- or microscale (Fig. 1c). For instance, to facilitate the transport of water droplets, each microfiber consists of a hygroscopically responsive alginate–diatomite shell embedded with an array of eccentrically distributed microparticles (Fig. 1b). Upon

contact with a water droplet, the hygroscopic swelling induces local deformation, giving rise to dynamically evolving geometric gradients. By introducing asymmetries—either in the droplet deposit position or in the surface microstructures (Fig. 1c)—these gradients can guide the droplet along programmable, steerable trajectories.

### Design realization

To implement the proposed design, we demonstrate a hygroscopically responsive SEMR capable of spontaneous water droplet transport (Fig. 2a). The constituent microfibers are fabricated using droplet microfluidic technology (Supplementary Fig. S1 and Supplementary Video 1), enabling precise control over their morphology by tuning the properties of the embedded microparticles (Supplementary Fig. S2). The eccentrically distributed microparticle arrays create a rugged side decorated with periodic micro-protrusions, while the encapsulating diatomite-alginate shell forms a flat spine on the opposite side (Fig. 2b). The alginate-diatomite material is hygroscopic, while the microparticles are unaffected by humidity changes. Upon contact with water droplets, the microfibers bend toward the microparticle side due to asymmetric volumetric changes caused by hydration (Fig. 1b)[46]. The degree of microfiber deformation and hydration can be tuned by modifying flow parameters and material composition (Supplementary Fig. S3) during fabrication.

To enable directional droplet transport, two parallel microfibers, each with a length $L_f$, are suspended in air with both ends fixed to a solid substrate (Fig. 2a). The microfibers' rugged sides face inward, with an initial spacing $s_0$. When a water droplet (~1.8 μL) is deposited onto the SEMR ($L_f = 25$ mm and $s_0 = 0.6$ mm), the transport process occurs in three stages: (I) bidirectional spreading, (II) asymmetric retraction, and (III) directional transport (Fig. 2a and c). The evolution of the droplet's two contact line positions characterizes the dynamics and shape changes across the three stages (Fig. 2c).

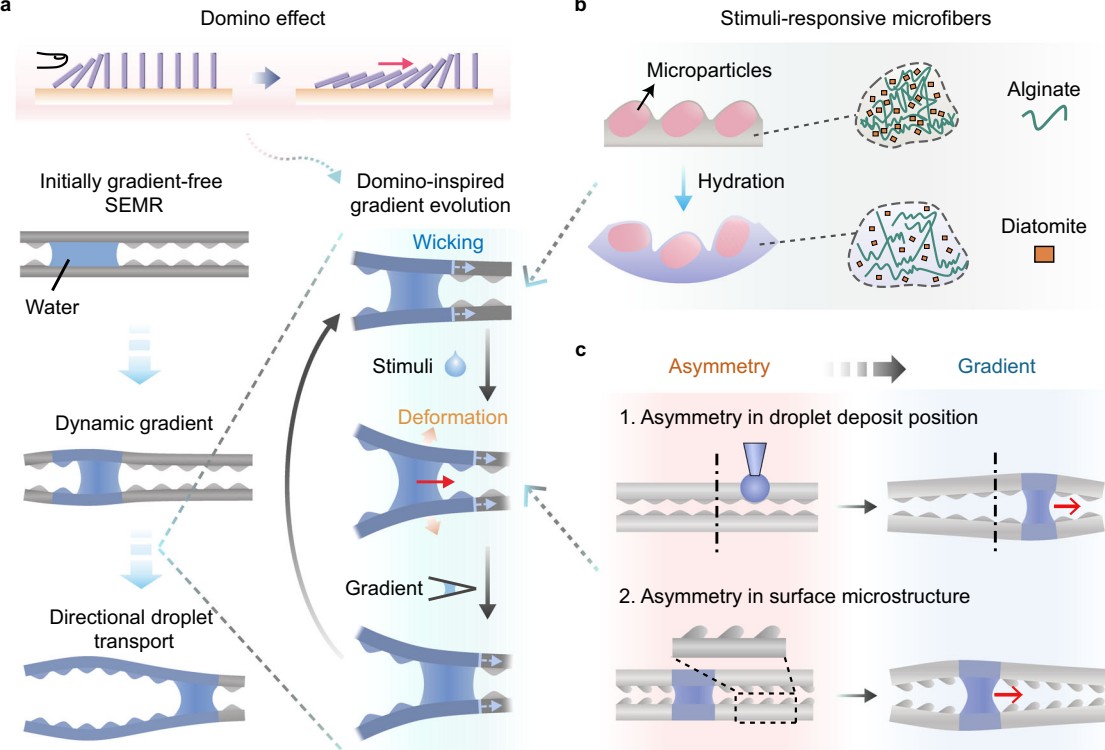

**Fig. 1 | Design of the shape-evolving microfiber rail (SEMR). a** Domino-effect-inspired design of a non-gradient SEMR system composed of two parallel microfibers with stimuli-responsive deformability. The system dynamically generates geometric gradients to drive directional transport of droplets. **b** Schematic showing the hygroscopically responsive deformation of the microfiber designed for directional water droplet transport. **c** Asymmetry in either the droplet deposit position or surface microstructures, laying the basis for generating dynamically evolving gradients.

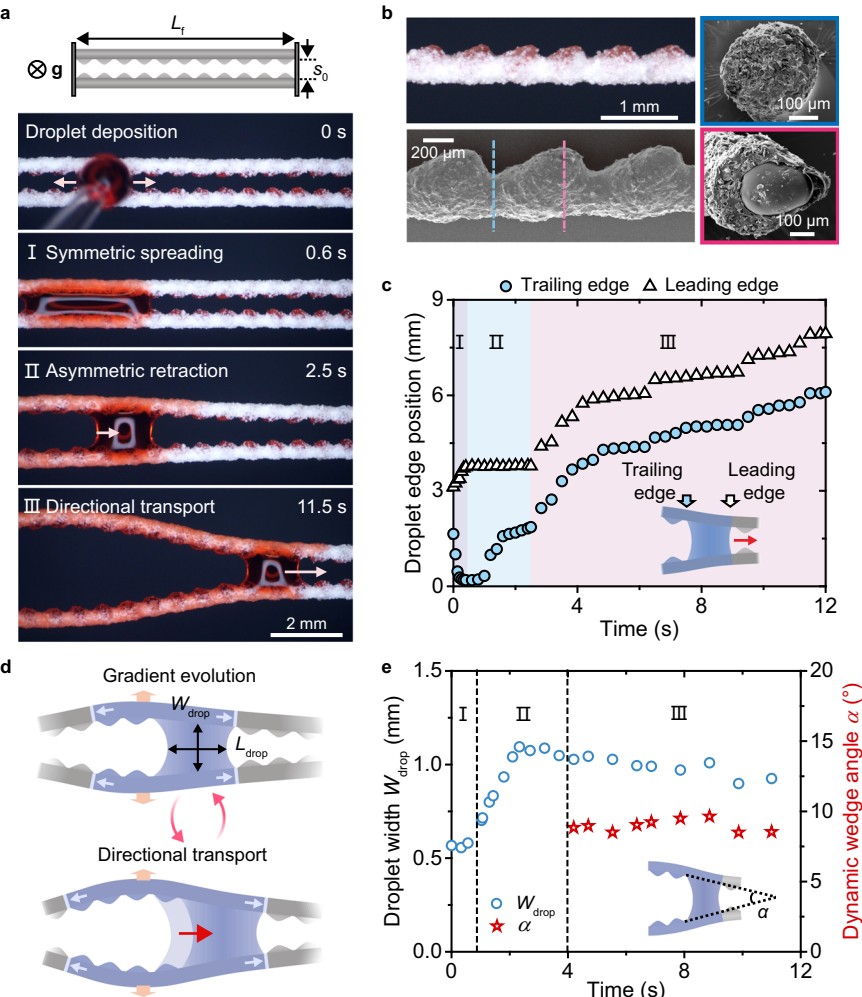

**Fig. 2 | Directional droplet transport on SEMR. a** Directional transport of a water droplet (1.8 μL, dyed red) on the SEMR, displaying three stages after droplet deposition: (I) bidirectional spreading, (II) asymmetric retraction, and (III) directional transport. The microfiber arrangement is perpendicular to gravity. $L_f \approx 25$ mm, $s_0 \approx 600$ μm. **b** Optical (top left) and scanning electron microscopy (SEM; top right and bottom) images showing the structural characteristics, surface morphology, and cross-sections of the microfiber. The encapsulated microparticles create protrusions (bottom right SEM image), and the diatomite-alginate joints form valleys (top right SEM image) on the microfiber. **c** Plot showing the positions of the droplet's leading and trailing ends over time. The difference between the two ends defines the droplet's body length. **d** Schematic showing droplet-microfiber interactions during directional droplet transport. The droplet's dimensions are characterized by its width ($W_{drop}$) and length ($L_{drop}$). **e** Plot showing the droplet's characteristic width ($W_{drop}$) and the SEMR's dynamic wedge angle ($\alpha$) over time.

In the initial stage, the droplet spreads bidirectionally along the SEMR, forming an elongated liquid column sandwiched between the two microfibers (stage I, 0 s to 0.6 s, Fig. 2a). The maximum spreading length of the droplet, $L_s$, scales with the droplet volume, $\Omega$ (Supplementary Fig. S4). This spreading occurs rapidly (within 0.6 s), during which the microfibers remain stiff, maintaining a nearly constant spacing.

Subsequently, the hydration-induced deformation of the SEMR transforms the parallel rail into a spindle-like configuration (stage II, 0.6 s to 2.5 s, Fig. 2a). While capillary suction from the droplet can cause coalescence between non-responsive microfibers, as observed in both previous studies[47] and our work (Supplementary Fig. S5), the hydration-driven deformation of the SEMR generates a force sufficient to counteract this effect (Supplementary Fig. S6 and Supplementary Note S1). The expansion of the spindle-like microfiber spacing facilitates droplet receding. During this process, the asymmetric retraction of the droplet propels the droplet preferentially toward one side of the spindle rail. The broken symmetry at this stage results in the droplet assuming a wedge-like shape, with one end larger and the other smaller, guided by the geometric configuration of the microfiber rail.

Following asymmetric retraction, a geometric gradient emerges as the SEMR adopts a wedge-shaped configuration. This shape preserves the droplet's asymmetry and drives its directional motion toward the narrower end, propelled by the Laplace pressure gradient (stage III, 2.5 s to 11.5 s, Fig. 2a). During this directional transport stage, the droplet's characteristic width ($W_{drop}$) and the SEMR's wedge angle ($\alpha$) remain nearly constant, in contrast to the spreading and receding stages (Fig. 2d, e). This stability reflects a coordinated interplay between microfiber deformation and droplet motion, facilitating steady and continuous transport.

The SEMR further demonstrates the ability to transport droplets against gravity (Fig. 3a), and functions effectively across a broad range of droplet surface tensions $\gamma$ (Fig. 3b and Supplementary Fig. S7a) and viscosities $\mu$ (Fig. 3c and Supplementary Fig. S7b). Upon dehydration, the SEMR passively returns to its original parallel, non-gradient configuration (Fig. 3d), resetting the system for subsequent droplet transport. The rate of this dehydration-driven reset is highly dependent on ambient temperature, with higher temperatures leading to faster recovery (Fig. 3e). After 100 hydration–dehydration cycles, the deformation performance remains robust (Fig. 3f), indicating great

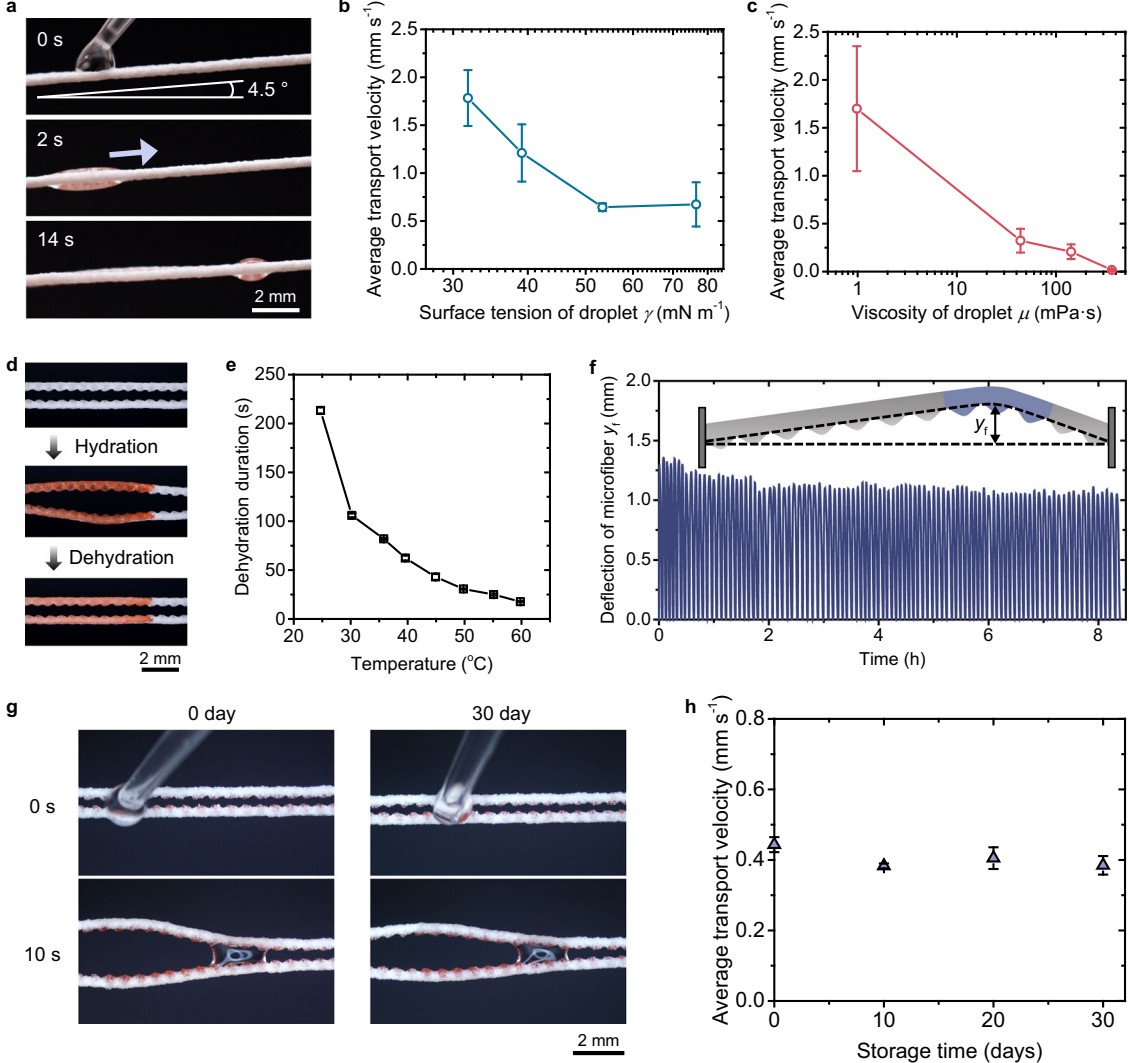

**Fig. 3 | Transport capability and durability of SEMR. a** Time-sequential images showing a 2 μL droplet ascending uphill on the SEMR tilted at 4.5°. **b, c** Effects of surface tension γ (**b**) and viscosity μ (**c**) on average transport velocity. **d** SEMR undergoing hydration to evolve dynamic gradients and subsequent dehydration to restore its initial gradient-free topology. **e** Effects of ambient temperature on dehydration duration of SEMR. **f** Deflection of microfibers during hydration and dehydration cycles. The deposited droplet volume is 0.4 μL. **g** Snapshots comparing the droplet transport performance of the SEMR after 30 days of storage. **h** Plot of average transport velocity as a function of storage time, demonstrating long-term performance stability. Error bars in (**b, c, e, h**) indicate the standard deviation for three measurements at each data point.

reversibility and mechanical resilience. Moreover, transport efficiency remains stable after 30 days of storage (Fig. 3g, h). While this study focuses on spindle-like deformation, the configuration of microfibers can be modified to tune SEMR's deformation behavior, enabling diverse liquid transport modes (Supplementary Fig. S8).

**Direction steering**

The asymmetric retraction of a droplet acts analogously to the initial domino in a chain reaction, setting the direction for subsequent droplet transport on the SEMR. A comprehensive understanding of the origins of this asymmetry during droplet retraction is essential for effectively guiding the direction of transport. We identify that both the macroscopic topology and microscopic structures of the SEMR can be strategically engineered to precisely control the transport direction (Fig. 1c).

In our first demonstration, we design and fabricate a SEMR (Fig. 4a) featuring symmetric microstructures on its rugged surface. These mound-like microstructures, formed by the encapsulated trimethylolpropane ethoxylate triacrylate (ETPTA) microparticles, exhibit nearly identical inclined angles on both sides ($\beta_1 \approx \beta_2 \approx 40°$). With this design, the direction of droplet transport depends on its deposit position, $\varepsilon$, defined as the ratio of the deposited distance $L_d$ (relative to the SEMR's left end) to the microfiber length $L_f$, $\varepsilon = L_d/L_f$ (Fig. 4a). When $\varepsilon < 0.5$, the droplet moves toward the left end (negative direction), whereas when $\varepsilon > 0.5$, it moves toward the right end (positive direction, Fig. 4b, c). This behavior indicates that the droplet consistently gravitates toward the nearest fixed end of the SEMR (Supplementary Video 2). Interestingly, when a droplet is deposited at the center ($\varepsilon = 0.5$), it can move either left or right, resulting in two potential outcomes, referred to as the transition point (Fig. 4c).

In this configuration, macroscopic topological asymmetry dominates the droplet's asymmetric retraction, steering directional transport. For instance, when $\varepsilon > 0.5$, the microfiber on the droplet's left side deflects with a higher speed than the right, due to the reduced mechanical confinement imposed by the left-side fixed substrate (Fig. 4d). This asymmetry causes the contact line on the left to retract

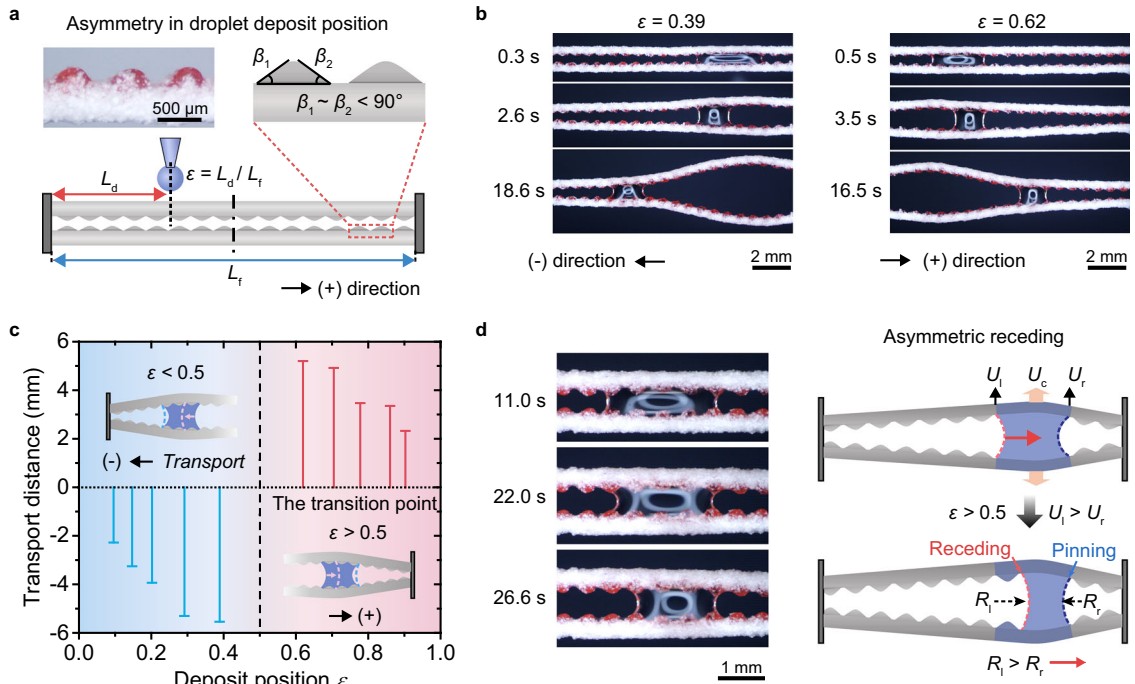

**Fig. 4 | Macroscopic topological asymmetry dominating selective directional droplet transport. a** Directional droplet transport on SEMR with symmetric microstructures. **b** Time-sequence images showing water droplet transport on SEMR in opposite directions depending on the droplet deposit position $\varepsilon$. When $\varepsilon = 0.39$, the droplet moves to the left (negative direction), while at $\varepsilon = 0.62$, it moves to the right (positive direction). **c** Dependence of the droplet transport distance on the droplet deposit position $\varepsilon$. As $\varepsilon$ increases from -0.1 to -0.9, the transport direction shifts from negative to positive, with the transitional position occurring at $\varepsilon = 0.5$. **d** Time-sequence images and schematic diagrams illustrating the asymmetric receding dynamics of droplets, governed by macroscopic topological asymmetry.

---

earlier, initiating directional motion. The deflection speed ratio between the left ($U_l$) and right ($U_r$) sides can be approximated as (see Supplementary Note S2),

$$\frac{U_l}{U_r} = \frac{1 - \frac{L_s}{2\varepsilon L_f}}{1 - \frac{L_s}{2(1-\varepsilon)L_f}} \qquad (1)$$

where $L_s$ is the maximum spreading length of the droplet. When $\varepsilon > 0.5$, the speed ratio exceeds unity ($U_l/U_r > 1$), resulting in a larger rail spacing and a greater radius of curvature on the left ($R_l > R_r$). For a completely wetting droplet confined between two angled fibers, the Laplace pressure difference ($\Delta P$) across the droplet is independent of the fiber diameter and given by (Supplementary Fig. S9 and Note S3),

$$\Delta P = \gamma \left( \frac{1}{R_r} - \frac{1}{R_l} \right) \qquad (2)$$

Given that $R_l > R_r$, the Laplace pressure difference is positive ($\Delta P > 0$), generating a net force from left to right and propelling the droplet in the positive direction. Conversely, when $\varepsilon < 0.5$, the force reverses, directing droplet transport to the left. In this way, the SEMR boundary effectively governs the direction of droplet transport.

To achieve more versatile control, we introduce another SEMR with asymmetric microstructures. These microfibers, encapsulating polydimethylsiloxane (PDMS) microparticles (Fig. 5a), exhibit ratchet-like features with two distinct inclined angles ($\beta_1 = 47.3° < 90° < \beta_2 = 110.8°$). While previous studies have utilized ratchet structures primarily to direct liquid film spreading[1,23], our approach leverages these asymmetric features to realize controlled directional transport of discrete droplets. As demonstrated in Fig. 5b and Supplementary Video 3, a droplet moves in the positive direction (aligned with the

ratchet tilt) even when $\varepsilon < 0.5$ (e.g., $\varepsilon = 0.38$). This finding highlights the dominance of asymmetric microstructures over macroscopic topology in determining transport direction. Systematic experiments reveal a significantly lowered transition point ($\varepsilon \approx 0.23$) for switching between negative and positive directions (Fig. 5c), compared to $\varepsilon = 0.5$ in the symmetric case. For $\varepsilon < 0.23$, the macroscopic boundary confinement induces transport in the negative direction, while for $\varepsilon > 0.23$, the asymmetric microstructures govern droplet retraction toward the positive direction.

This behavior can be explained by localized force analysis at the droplet's contact lines. For fully wetted microfibers with a diatomite-alginate shell, the water contact angle approaches zero (Supplementary Fig. S10), rendering the capillary force tangential to the microstructure profile. During contact line retraction over the ratchet, the capillary force evolves dynamically (Fig. 5d). For instance, as the contact line retracts left-to-right (positive direction) over the left ratchet apex (position a −), the horizontal capillary force component ($F_{h-}$) decreases from $F_{h-} = \gamma$ to $F_{h-} = \gamma\cos\beta_2$, while a downward vertical component ($F_{v-} = \gamma\sin\beta_2$) emerges, aiding movement. Conversely, during retraction right-to-left over the right ratchet apex (position a +), $F_{h+}$ decreases to $\gamma\cos\beta_1$ and the generated $F_{v+} = \gamma\sin\beta_1$ points upward, opposing retraction.

This asymmetric force distribution induces faster contact line motion over the left ratchet apex, while pinning occurs at the right apex, displacing the droplet's centroid asymmetrically to the right. A substantial difference between the two inclined angles, with $\beta_1 < 90° < \beta_2$, is essential to trigger such asymmetric receding dynamics. In this case, downward-directed force component $F_{v-}$ facilitates the descent of the contact line along the micro-protrusions on the obtuse-angle side ($\beta_2$), whereas the upward-directed force component $F_{v+}$ on the acute-angle side ($\beta_1$) suppresses contact line descent. Consequently, ultra-fast receding of the contact line occurs along the $\beta_2$ side (Fig. 5d

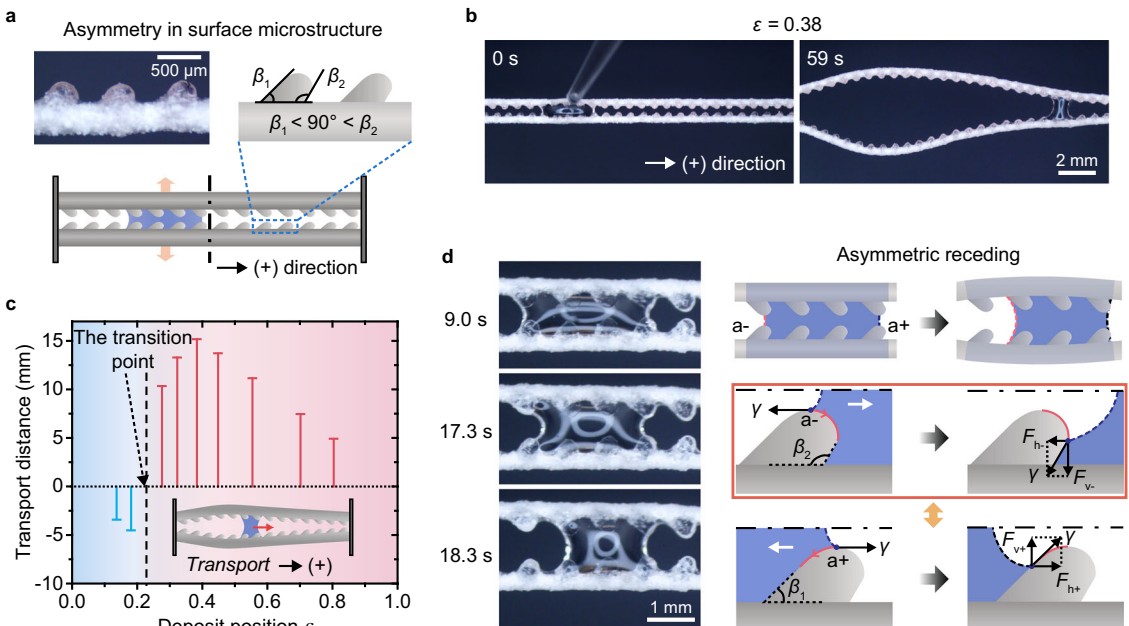

**Fig. 5 | Asymmetric microstructures dominating selective directional droplet transport. a** Directional droplet transport on SEMR with asymmetric micro-structures. **b** Time-sequence images of water droplet transport unidirectionally in the positive direction on SEMR, even when $\varepsilon = 0.38$. **c** Variation in droplet transport distance as $\varepsilon$ changes, with the transitional point from negative to positive transport occurring at $\varepsilon \approx 0.23$. **d** Time-sequence images and schematic diagrams depicting the asymmetric receding dynamics of droplets, governed by asymmetric microstructures.

and Supplementary Video 4) immediately after crossing the left ratchet apex (position a−). This hypothesis is further validated by fabricating SEMRs with both $\beta_1$ and $\beta_2$ less than 90°, causing both $F_{v-}$ and $F_{v+}$ to point upward (Supplementary Fig. S11a), thereby inhibiting contact line retraction. Under these conditions, variations in the microstructure asymmetry ($\beta_1/\beta_2$) exhibit negligible impact on droplet transport dynamics, including directionality and velocity (Supplementary Fig. S11b, c).

**Droplet transport by dynamic gradient**

The directional droplet transport is governed by water adsorption within the microfiber body, which induces deformation and generates a dynamically evolving geometric gradient that propels the droplet (Fig. 6a). A remarkable feature of the SEMR is its unique independent control of transport velocity and distance. The average transport velocity remains nearly constant regardless of the transport distance, provided the droplet volume is fixed (Fig. 6b). Unlike traditional systems reliant on static gradients, SEMR's dynamic response demonstrates adaptive capability by segmenting the movement path into cascading gradients. This adaptive mechanism allows variations in transport distance to depend on the number of cascades rather than the magnitude of the gradient, enabling the decoupling.

Wicking plays a critical role in sustaining the interactions between droplets and microfibers. Contact angle hysteresis impedes droplet movement on dehydrated microfibers. However, water wicking along the microfiber counteracts this by forming a lubricating liquid film and hydrating the microfiber to facilitate deformation (Fig. 6c and Supplementary Fig. S12). When the wicking process is limited, droplet transport becomes stepwise, exhibiting motion-stop cycles where the transport velocity alternates between peaks and pauses (Fig. 6d). This behavior resembles a chain reaction of falling dominoes, with the droplet pausing until the wicking-induced dynamic geometric gradient generates sufficient Laplace pressure to resume movement. This cyclical mechanism develops dynamic gradients that drive subsequent transport cycles.

The competition between wicking and microfiber deformation determines the droplet behavior. Insufficient wicking relative to deformation leads to excessive spacing between the microfiber rails, forcing the droplet to expand in width ($W_{drop}$). This increases the width-to-length aspect ratio, $C = W_{drop}/L_{drop}$, where $L_{drop}$ is the droplet length (Fig. 2d). Once $C$ exceeds a critical value, $C_{criti} \approx 2.41$ (Supplementary Fig. S13), the droplet spontaneously breaks apart due to Rayleigh-Plateau instability (Fig. 6e). Consequently, the droplet's fate on SEMR—directional transport or breakup—is dictated by the relative timescales of liquid wicking and microfiber deformation.

The characteristic wicking time ($t_w$) is defined as the time required for water to diffuse over a characteristic length along the microfiber (Supplementary Fig. S14). It scales as

$$t_w \sim \frac{L_c^{\,2}}{D} \tag{3}$$

where $L_c$ denotes the characteristic distance between two adjacent protrusions on the microfiber, and $D$ is the water diffusion coefficient along the fiber (see Supplementary Fig. S3c, d). The characteristic deformation time ($t_d$) represents the interval from the onset of microfiber deformation to droplet breakup, approximated by

$$t_d \sim \frac{\left(\Omega C_{criti}^{\,2}\right)^{\frac{1}{3}} - s_0}{U_c} \tag{4}$$

where $U_c$ is the microfiber separation speed (Supplementary Note S4). A linear correlation between $t_w$ and $t_d$ delineates the boundary between transport and breakup regimes in the phase diagram (Fig. 6f). Directional transport occurs when $t_d > t_w$, indicating that wicking must outpace deformation to prevent droplet breakup. Enhancing the wicking process, such as by increasing the diffusion coefficient $D$, significantly improves transport speed, enabling continuous droplet motion (Fig. 6g) while preserving the characteristic step-wise behavior in transport velocity (Supplementary Fig. S15). Environmental factors

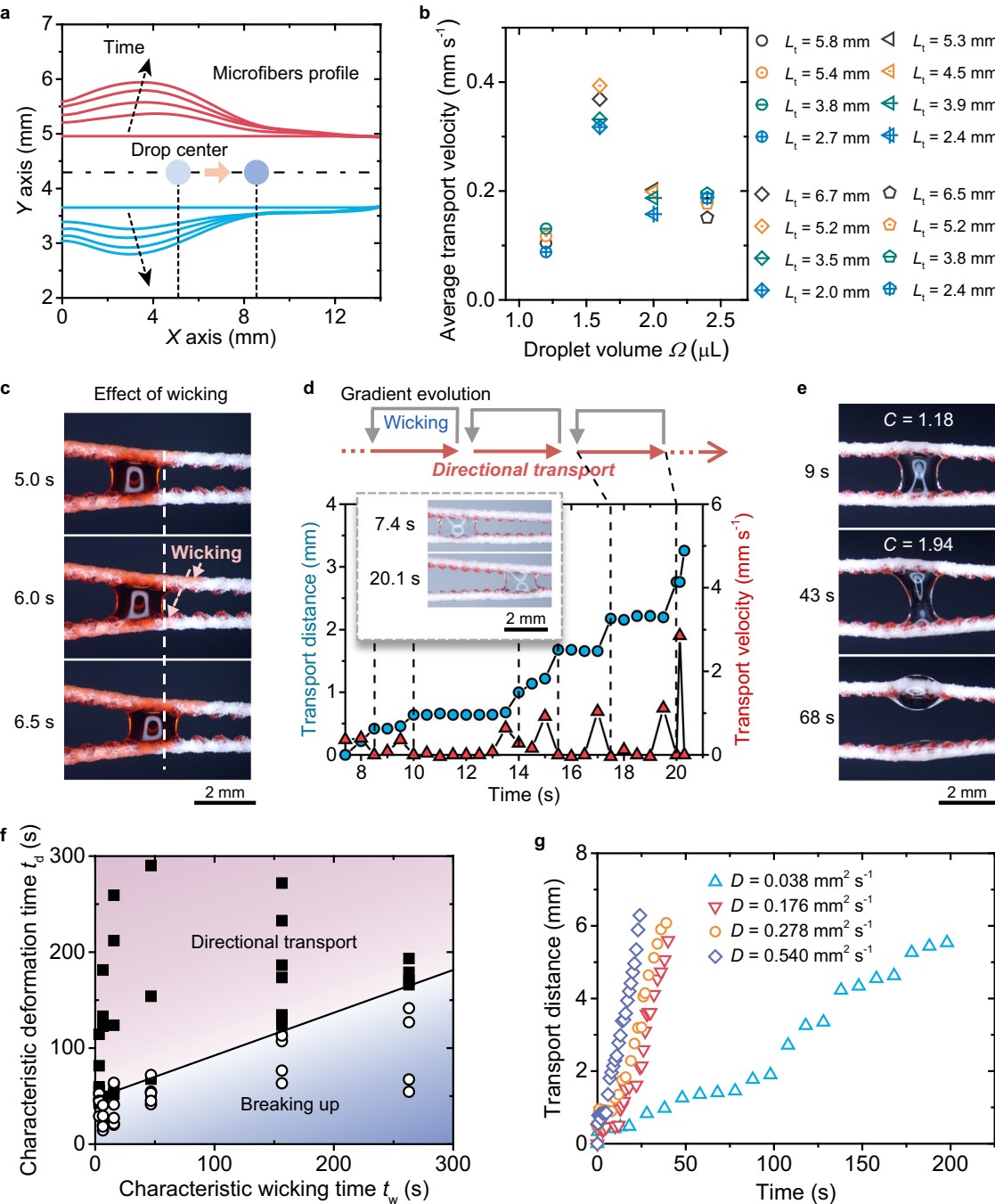

**Fig. 6 | Dynamics of directional droplet transport with droplet-surface interactions. a** Dynamic evolution of the microfibers profile during droplet transport on SEMR. **b** Independent control over the average transport velocity and distances $L_t$ for droplets with different volumes $\Omega$. **c** Time-sequence images showing the wicking effects on the surface of SEMR during droplet transport. **d** Variations in droplet transport distance and velocity versus time. **e** Breaking up of a droplet trapped between two microfibers due to an increasing width-to-length aspect ratio $C$. **f** Phase diagram showing two droplet behaviors—directional transport and breaking up—depending on the characteristic wicking time $t_w$ and the deformation time $t_d$. **g** Transport distance versus time on SEMR with varied diffusion coefficient $D$. The droplet volume is 2 μL.

such as humidity and temperature also affect microfiber deformation and droplet transport dynamics (Supplementary Fig. S16), as they influence the wetting properties of the microfiber surface.

Although water adsorption and liquid film formation can induce liquid loss during transport (Supplementary Fig. S17), lubricated SEMRs markedly enhance droplet transport performance. Prewetting has previously been employed to boost droplet velocity on dry surfaces[36,48], and a similar effect is observed here. Following the passage of a single droplet, the initially dry SEMR becomes lubricated and

deformed, enabling the continuous transport of subsequent droplets at velocities up to ~15 mm s⁻¹ with water loss reduced to ~10% (Fig. 7a, b). Furthermore, the liquid loss can be significantly minimized—to approximately 0.01 μL mm⁻¹ (volume per unit transport distance)—by coating part of the SEMR with a non-hygroscopic material (Fig. 7c). This modification simultaneously preserves the hygroscopically induced deformation required for gradient evolution while providing a non-hygroscopic property for long-distance transport of minute droplets. Compared with the uncoated SEMR, which transports microliter-

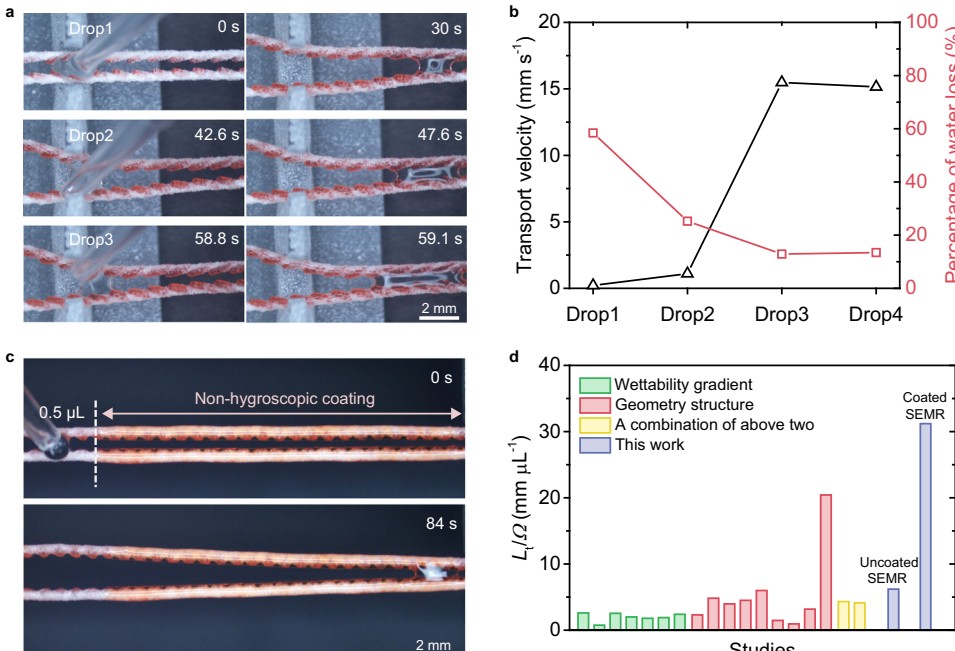

**Fig. 7 | The versatility of SEMRs. a** Time-sequence images showing the continuous transport of multiple drops (2 µL) along the SEMR. **b** Variations in droplet transport velocity and water loss versus the number of droplets transported. **c** Optical images showing the transport of a 0.5 µL water droplet along the coated SEMR. In the upper image, the dashed line marks the boundary between regions with (right) and without (left) the non-hygroscopic coating. **d** Comparison of transport distance per droplet volume ($L_t/\Omega$) among surfaces without external energy input. The green, red, and yellow columns indicate the reported transport performances for surfaces with wettability gradients, geometry structures, and their combinations, respectively (Supplementary Fig. S19).

scale droplets (Supplementary Fig. S18 and Supplementary Video 5), the coated SEMR enables reliable nanoliter droplet transport (Fig. 7c) and achieves a substantially higher transport distance per droplet volume (Fig. 7d and Supplementary Fig. S19). These results highlight the high material compatibility and design flexibility of SEMRs, demonstrating that the transport performance can be finely tuned through simple surface modifications.

### Potential applications

SEMR shows great promise for applications in diverse fields, including analytical chemistry, cargo transport, electronic circuits, and medical diagnostics. A proof-of-concept demonstration illustrates the potential of SEMR in droplet manipulation. When multiple droplets are placed on SEMR at specified intervals, the rear droplet (Droplet II) accelerates to merge with the leading droplet (Droplet I) once the microfiber surface between them becomes fully wetted (Fig. 8a–c, and Supplementary Video 6). Notably, a peak transport velocity of ~21 mm s$^{-1}$ is attained during this droplet-chasing process. This characteristic enables SEMR to effectively suspend droplet microreactors, such as successfully manipulating the coalescence of KCl and AgNO$_3$ droplets to trigger a chemical reaction (Fig. 8d).

Beyond manipulating droplets that interact directly with microfiber rails, SEMR also facilitates the directional transport of non-interactive materials, such as oil droplets and solid cargos. In the case of oil droplets, water droplets act as "fuel", inducing microfiber deformation that creates a geometric gradient to propel the oil droplets away from the water droplets. This process accommodates a broad viscosity range of oil droplets (80–3000 cSt, as shown in Fig. 8e and Supplementary Video 7). For solid cargos, such as PDMS spheres, water droplets function as robotic carriers, driving directional transport while gradually depleting in volume during the unloading process (Fig. 8f and Supplementary Video 8). This multiphase manipulation capability is preserved even at smaller scales. For instance, SEMRs facilitate oil-phase enrichment via controlled water extraction from

oil-in-water emulsions and enable uniform deposition of microparticles on microfiber surfaces following the transport of a suspension droplet (Supplementary Fig. S20).

We further highlight applications that leverage the interaction between aqueous droplets and the SEMR surface. For instance, water droplets can bridge disconnected circuits, creating a droplet-based input/output (I/O) port system. In this configuration, two light-emitting diodes (LEDs) are connected to the left and right ends of the SEMR. By controlling the initial release position of a water droplet (input signal), the system selectively activates LEDs on different sides (output) depending on the droplet's transport direction (Fig. 8g and Supplementary Video 9).

In diagnostic applications, SEMR can be preloaded with colorimetric indicators to analyze droplet composition through chemical reactions triggered by liquid wetting. For example, when a 2 µL citric acid droplet traverses a SEMR pretreated with cresol red and sodium hydroxide (NaOH), the microfiber surface transitions from purple to yellow, indicating the presence of acidic components in the droplet (Fig. 8h and Supplementary Video 10). This capability allows for high-sensitivity detection with much smaller sample volumes compared to conventional diagnostic devices, presenting great potential for applications in biomedical diagnostics with diverse biological samples, such as blood, saliva, and urine.

### Discussion

We have demonstrated spontaneous and directional droplet transport on SEMRs, enabled by dynamically evolving geometric gradients that result from microfiber upon droplet–surface interactions. In contrast to traditional surfaces with pre-defined static gradients, SEMRs operate as initially non-gradient systems and leverage a domino-inspired design to trigger cascading gradients, enabling continuous and long-range droplet transport. This dynamically-evolving mechanism allows for on-demand and independent tuning of transport direction, velocity, and distance, thereby overcoming the inherent limitations of

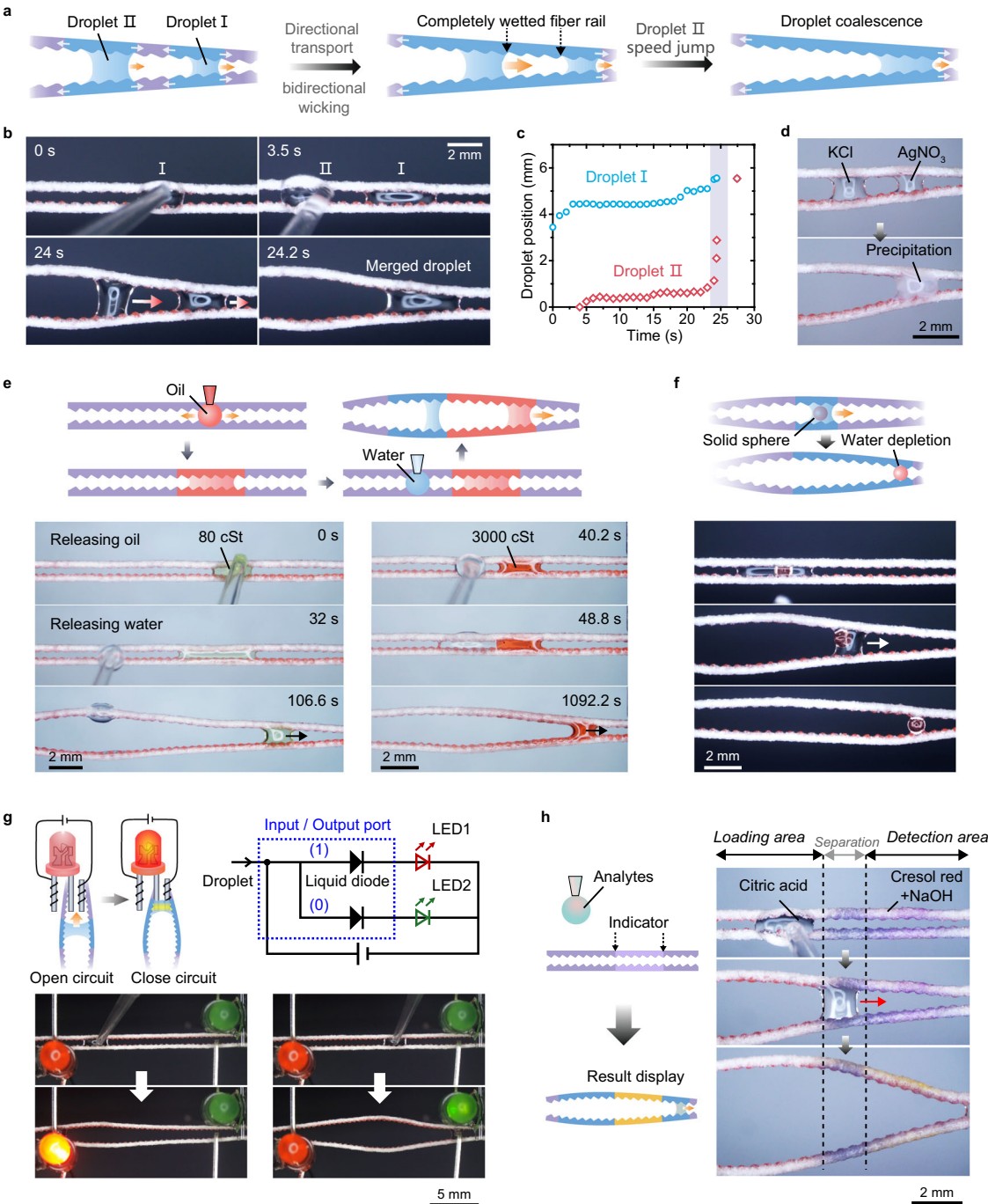

**Fig. 8 | Applications of directional droplet transport on SEMRs. a, b** Schematic (**a**) and photographs (**b**) showing spontaneous merge of droplets on SEMR. **c** Plots of the moving distance of droplets I and II in (**b**) over time, showing the speed jump of droplet II before merging. **d** Time-sequence images demonstrating a droplet-based micro-reactor on SEMR. **e, f** The directional transport of an oil droplet (**e**) and a solid sphere (**f**) enabled by water droplets. **g** Intelligent fluidic-electronic circuits enabled by SEMR. Selective directional transport of water droplets on SEMR can light up different light-emitting diodes (red and green). **h** Component detection by droplet-microfiber interaction on SEMR. After droplet transport, the microfiber turns from purple to yellow, indicating the presence of acidic substances in the droplet.

conventional static-gradient surfaces. Moreover, the transport performance can be further enhanced through surface modifications that tailor droplet–surface interactions. By leveraging these distinctive features, SEMR has proven versatile in applications such as micro-reactors, solid/liquid manipulation, liquid circuits, and material detection. The dynamic interaction-based design of SEMRs establishes a foundation for next-generation platforms in intelligent liquid manipulation, offering significant opportunities for advanced fluidic technologies.

## Methods
### Materials
Sodium alginate (Aladdin), diatomite (Kieselguhr from Sinopharm Chemical Reagent Co., Ltd.), calcium chloride ($CaCl_2$, Aladdin), poly(-vinyl alcohol) (PVA, 87–89% hydrolyzed, Aladdin), distilled water (Watsons), ETPTA (Dieckmann), PDMS (Dow), silicone oil (viscosity of 0.65, 80, and 3000 mPa·s, Aladdin), hydrophilic nano fumed silica (7–40 nm, Aladdin), $AgNO_3$ (Acros), KCl (Aladdin), NaOH (Dieckmann), citric acid (AR, Dieckmann), cresol red (Aladdin), sodium dodecyl

sulfate (SDS, Macklin), carboxymethyl cellulose (CMC, Aladdin), ethanol (absolute, Anaqua), trimethoxy-[3-(2-methoxyethoxy)propyl] silane (TM-MES, Macklin), UV curable resin (Ergo 8500, Kisling) and fluorescent particles (1–2 μm, Btfluid) were used as purchased.

## Fabrication of microfluidic devices

Co-flow microfluidic devices were fabricated by inserting an inner glass capillary with a tapered tip into a cylindrical outer one with a larger diameter. This configuration enabled the generation of oil-in-water (O/W) emulsions. The outlet of the outer capillary was inserted into a Petri dish for microfiber cross-linking.

## Fabrication of the microfiber

The fabrication process of the microfiber is depicted in Supplementary Fig. S1. For the preparation of SEMR with symmetric microstructures, the internal phase was photocurable ETPTA containing 2 wt% photo-initiator, while the external phase consisted of a mixture of sodium alginate (5 wt%) and diatomite (1 wt%–5 wt%). Both phases were pumped into a microfluidic device using two syringe pumps (Longer Pump, LSP01-1A) to independently control the flow rates. The resulting emulsion was then directed into a Petri dish containing a 25 wt% CaCl$_2$ aqueous solution, triggering rapid cross-linking of the sodium alginate fiber shell. Subsequent UV irradiation solidified the internal ETPTA microdroplets into microparticles. For the preparation of SEMR with asymmetric microstructures, the internal phase was a mixture of PDMS and silicone oil (0.65 cSt) in a 1:1 mass ratio. After fiber cross-linking, the microfibers were placed in an 80 °C water bath for two hours to cure the internal PDMS microparticles. The microfibers were then washed five times to remove residual CaCl$_2$ and gradually straightened during the drying process.

## Enhancing the wicking capability of the microfiber

The microfibers were first immersed in a 2 wt% CaCl$_2$ aqueous solution for 5 s, then removed and air-dried before being straightened. Following this, the microfibers were immersed in an alginate solution (1 wt%) uniformly dispersed with hydrophilic silica (0.01–0.05 wt%), and after 3 s, they were removed, forming a sodium alginate-silica coating on the surface. The microfibers were then washed five times and air-dried. By carefully controlling the small amount of alginate, the resulting coating layer remained sufficiently thin, thereby preserving the microfibers' deformation capability.

## Non-hygroscopic coating of SEMRs

The microfluidically-generated microfibers were uniformly coated with a photocurable resin, followed by UV curing for 30 s. Subsequently, they were modified with TM-MES and left undisturbed for 24 h.

## SEMRs for pH detection

A 0.2 μL water droplet containing cresol red (2.47 mM) and NaOH (2.65 mM) was deposited on the SEMR. After the microfibers naturally dried, the SEMR was used for pH detection, with the purple area indicating the detection zone.

## Control of humidity

SEMRs were placed inside a semi-enclosed glass chamber, where the ambient humidity was precisely regulated using a combination of a humidifier (YADU, YC-D205) and a dehumidifier (Midea, CF12BD/N3-OQ1). The environmental humidity was continuously monitored with a digital humidity sensor (Accurate, TH10R) to ensure stable and accurate control.

## Control of temperature

To regulate temperature, SEMRs were suspended above a hot plate (JFTOOLS, JF-976B), allowing uniform thermal exposure. The temperature was continuously monitored using a thermocouple positioned on the SEMR to ensure precise thermal control throughout the experiments.

## Characterization

Microstructures of SEMRs were characterized by scanning electron microscopy (SEM, FEI Quanta 450 FEG). Optical microscopy images and videos were captured by a stereomicroscope (SOPTOP SZM7045) equipped with a camera (Harmony Technology HK Ltd., 20 MSPM). Contact angles were measured by a contact angle goniometer (SINDIN, CSCIDC-350). The fluorescent particles are excited by the laser (FULEI, FU520AB1800, wavelength of 520 nm). Images were analyzed using ImageJ.

## Data availability

The data that support the findings of this study are available from the corresponding author. Source data are provided with this paper.

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

## Acknowledgements

The financial support from the National Natural Science Foundation of China (52303046 and 12388101), the Research Grants Council of Hong Kong (21213621 and R1017-24F), Shenzhen Science and Technology Program (JCYJ20220530140812028), and the City University of Hong Kong (7006097) is gratefully acknowledged. Open Access made possible with partial support from the Open Access Publishing Fund of the City University of Hong Kong.

## Author contributions

P.Z. conceived the research. P.Z. and S.W. designed the experiments. Y.Z., W.Z., Y.L., and S.T. assisted in microfiber fabrication using the microfluidic method. S.W. conducted the experiments. P.Z., S.W., and T.S. performed the theoretical analysis. P.Z. and S.W. analyzed the data and wrote the manuscript. T.S. reviewed and provided insightful suggestions on the manuscript. P.Z. supervised the research.

## Competing interests

The authors declare no competing interests.
