## [Transparent Peer Review file · Nature Communications]

Spontaneous droplet transport on shape-evolving microfiber rails

Corresponding Author: Dr Pingan Zhu

Version 0:

Reviewer comments:

Reviewer #1

(Remarks to the Author)

The self-adaptive microfiber rails (SAMRs) represent a novel approach to directional droplet transport, offering significant advancements over conventional systems. The experimental method proposed is absolutely elegant and worthy of publication. The way the manuscript is written, and the lucid description of the setup is truly commendable. However, as the manuscript is up for a journal of repute like Nature Communications, I must analyze the rigor of the study and seek responses to further improvement of the manuscript. Below are certain issues/points that need to be addressed

The fabrication process relies on specific hygroscopic materials (e.g., diatomite-alginate shells), which may limit scalability or compatibility with other substances. Are these rails reusable? The six-cycle reusability is demonstrated. However, the transport dynamics are influenced by factors like humidity and temperature, which could impact performance in uncontrolled environments due to the impact of the ambient conditions on wettability and hysteresis.

The intricate design involving asymmetric microstructures requires precise control during manufacturing, which could increase production costs and complexity. For example, the $\beta=45^\circ$ in the illustrated case. Is it possible to obtain this level of precision?

Focussing on Fig 1d) we see the rails, after the passage of the droplets, keep on deforming to a very large extent. Excessive spacing between microfiber rails during deformation can lead to droplet breakup due to Rayleigh-Plateau instability when the width-to-diameter aspect ratio exceeds a critical value.

Further, the same figure shows a significant loss of droplet volume by wicking. In practical uses, the droplet would lose a lot of material in the arrangement. What would be the effective throughput as a function of ϵ

Focusing on Fig. 2, it appears that the rails tend to shrink prior to the droplet reaching the point. Is it because of the strain in the wake of the droplet (behind the droplet)? The droplet creating upstream deformation seems to be non-causal.

Figure 3 h is unclear. The deformation time is likely to be higher than the wicking time. That is not seemingly true for all the points.

Reviewer #2

(Remarks to the Author)

The present manuscript proposes an experimental investigation and proof-of-concept of a novel design for the directional transport of microfluidic droplets. In contrast with a permanent inhomogeneous geometry of the droplet's environment to drive its steady motion through surface energy minimization, the present process which relies on the use of flexible walls with complex macrostructures involving micro particle arrays and a hygroscopic shell that tends to absorb some of the droplet liquid and react to this absorption by asymmetric deformation. This wicking-induced deformation leads to confinement gradients or differential contact angle around the droplet, leading to the emergence of a driving force originating in this resulting asymmetric geometry.

The study is interesting and novel, the manuscript presents convincing details and validations of the concept. Beyond one major element that would absolutely need to be addressed by the authors before any decision regarding publication can be made, my comments are mostly minor and hopefully will help the authors enhance the clarity and pedagogy of their manuscript.

My most serious concern lies in some of the strong claims of the authors regarding the « self-adaptative » or « intelligent » nature of the material, or in the absence of steady gradients in their system. Admittedly, in its initial state, the microfluidic system only presents asymmetry at the microstructure level (if even) and reacts to the droplet presence by a deformation process that creates the necessary spatial gradient in an « unsteady » way. However, one should also argue that the gradient, despite invisible in the base state, remains nonetheless permanently encoded in the design of the system: repeating the experiment with different droplets seem to result in the same behavior, while one might have expected an « intelligent » system to « adapt » its reaction to the specific droplet or another form of non-unique stimulus. It would have been interesting for example to see if the system could be exploited for droplet sorting based on the droplet sizes, and its sensitivity. Also, the driving mechanism can still be associated with a « steady » gradient in free energy due to the irreversible transfer of liquid material in the wicking process.

Other detailed comments on the manuscript:

- this is probably the result of « fitting » within the Nature Communications format, but I have had a very hard time reading the figures that include too many panels and use exceptionally small fonts making it difficult to decipher the information (and there is a lot on each figure). I would recommend a better hierarchization of the information, leading to (i) a careful selection of some panels that have a more minor role and could either be omitted or transmitted to the SI, (ii) the division of some of the most crowded figures (e.g. Figure 1) into distinct figures and (iii) more details on the figure panels once they have been increased in size — in particular the explaining schematics that play a critical role for the reader's understanding.

- Introduction: The authors do not precise what they mean by « trade-off between speed and distance » nor references or studies where such trade off was discussed or reported.

- In some instances, in an effort to write the article in a catching and concise manner, some info needed to understand the meaning of the text is delayed to a later paragraph without any warning. For example, regarding the origin of the « asymmetry » mentioned around line 95: one only understands later on that the origin can take different forms: proximity to the attachment point of the fibers, microstructure asymmetry, etc...)

- As discussed in my major comment, the origin of the asymmetry, despite the claim of the authors, is intrinsically encoded in the system's design (through the asymmetric protrusions or the asymmetric distance to the bounding ends). How is that fundamentally-different from the classical « steady » gradient encoded in a microfluidic rail or anchor? The asymmetry just appears somewhat hidden temporarily and simply « revealed » but not « generated » by the droplet.

- A step critical to the robust functioning of the device is scarcely — if at all — described: dehydration. This is the step that will restore the system to its original state after the droplet has wicked it and gone by. One could interpret this as a « reloading » of the substrate's free energy so that it could drive a new droplet by monotonous energy transfer again. How is it achieved?

- Overall, I do not find the discussion of the timing aspect clear or convincing: on page 5, line 151, why does one side deflects « faster » (timing) than the other and not simply « more » (amplitude) ? What sets the timing?

- Line 182: the authors claim that ratchet geometries only facilitate liquid film spreading. What about Leidenfrost droplet propulsion on ratchet? Couldn't this be interpreted as direction droplet transport?

- page 7-8: I am confused by the discussion regarding the differential behavior of the two menisci on asymmetric protrusion. The current discussion suggests that both sides of the droplet act in the same direction regardless of the relative position of the two interfaces with respect to the protrusion pattern. I understand that the asymmetric microstructure may result for a single meniscus to create a horizontal force that is predominantly in one direction; yet I would anticipate that for some relative position in the protrusion and contact line, the direction of the forcing would be of opposite direction. Then, depending on the droplet size and relative position of the two sides, can there be a geometry-associated equilibrium?

Reviewer #3

(Remarks to the Author)

This manuscript proposes an innovative strategy of dynamically constructing structural gradients through asymmetric swelling of microfibers, theoretically overcoming the inherent trade-off between droplet transport velocity and distance in conventional intrinsic gradient systems. While the concept holds significant academic merit, several critical issues require substantial improvement:

1. Discrepancy Between Theoretical Promise and Experimental Performance. The reported droplet transport distance (centimeter-level) and velocity (millimeter/second scale) are notably inferior to state-of-the-art microfluidic systems. Dynamic contact line pinning induced by fiber surface roughness may be the primary limiting factor.

2. Oversimplified Laplace Pressure Analysis. The current single-curvature (RI) approach contravenes the fundamental Young-Laplace equation requiring orthogonal curvature components. Authors should: Develop mathematical models correlating RI and Rr with microfiber deformation and contact angle (θ).

3. Neglect of Fluid Property Effects. Surface tension (γ) and viscosity (η) as key parameters remain unexplored.

4. The discussion in Fig. 2e-f appears disconnected from the broader narrative of the manuscript. It is recommended that the authors repurpose this section to elucidate how the morphological features of SAMR in the Fig. 2a model (e.g., β_1 , the deflection speed ratio) govern liquid transport dynamics. A parametric study correlating these structural parameters with droplet acceleration (dv/dt) and Laplace pressure gradients would strengthen the mechanistic interpretation.

5. Quantitative structure-performance relationships Insufficient Reliability Validation Merely. 6 cycling tests inadequately demonstrate robustness. Required enhancements: I. 100-cycle positional drift statistics using optical positioning. II. Swelling reproducibility under varying humidity/temperature. III. 30-day storage performance degradation assessment.

6. The liquid transport mechanism driven by swelling-induced deformation inherently introduces liquid loss during the transport process. The authors should address this issue in the discussion and implement necessary optimizations to the fibers, rather than intentionally avoiding it.

7. The authors propose several applications based on this model in the manuscript, but these applications lack innovation,

and their practicality is questionable.

Recommendation

Major revision required. The work possesses transformative potential but currently falls short of Nature Communications' standards for mechanistic depth and technological impact.

Version 1:

Reviewer comments:

Reviewer #1

(Remarks to the Author)

I am very satisfied with the modifications and believe that the manuscript is fit for publication now. I would again like to emphasize on the elegance and simplicity of the methodology and believe the manuscript will be a worth archival material for Nat Comm.

Reviewer #2

(Remarks to the Author)

I thank the authors for taking the time and care to carefully address and respond to my comments on their original submission. I believe these changes strongly enhance the manuscript's clarity and impact and am therefore happy to recommend the publication of this manuscript.

Reviewer #3

(Remarks to the Author)

While the revised manuscript represents a significant improvement over the previous version, with notably enhanced figures and text, as well as substantial additional data supporting the claims, I regret that I cannot recommend publication in Nature Communications at this time. My primary concern relates to the core innovation claimed – namely, overcoming the inherent trade-off between droplet transport velocity and distance in conventional intrinsic gradient systems without external energy input – which motivated my previous recommendation for major revision.

However, the optimized performance metrics presented in this revision, specifically the achieved centimeter-level transport distance and millimeter-per-second velocity, do not convincingly demonstrate the claimed breakthrough in decoupling distance and velocity. This assessment is based on several key points: the supplementary video (S18) appears to contradict the stated transport distance, as measurement via the provided scale bar indicates droplet movement of approximately 7 mm over 17.4 seconds rather than the reported 12 mm, leading to transport speed overestimation.

Furthermore, comparison with prior art is insufficient and lacks rigor; crucially, multiple studies achieving droplet transport without external energy input report significantly superior performance, such as *J. Mater. Chem. A*, 2023, 11, 10164–10173 and *ACS Appl. Mater. Interfaces* 2017, 9, 9213–9220 (exceeding results by 1-2 orders of magnitude), and *Adv. Funct. Mater.* 2023, 33, 2212217 (analogous dual-fiber system showing several-fold higher metrics).

Additionally, the mechanism's critical reliance on droplet-induced fiber swelling introduces unaddressed limitations: small droplets may deplete due to absorption/swelling during long-distance transport; fiber recovery via evaporation imposes fundamental constraints on continuous/batch transport; and droplet size sensitivity may preclude practical applicability. These newly identified limitations concerning efficiency, operational continuity, and droplet size compatibility represent significant drawbacks directly impacting the claimed novelty and utility of the SEMR approach.

Therefore, based on unresolved concerns regarding performance benchmarks, data accuracy, and intrinsic system limitations, I conclude the manuscript in its current form does not meet Nature Communications' high-impact threshold.

Version 2:

Reviewer comments:

Reviewer #3

(Remarks to the Author)

My thanks to the authors for their careful revisions in response to my comments. Their revisions have made the manuscript stronger and more polished, enhancing both its clarity and overall impact. I am happy to recommend it for acceptance.

Response to the Comments by Reviewer #1

The self-adaptive microfiber rails (SAMRs) represent a novel approach to directional droplet transport, offering significant advancements over conventional systems. The experimental method proposed is absolutely elegant and worthy of publication. The way the manuscript is written, and the lucid description of the setup is truly commendable. However, as the manuscript is up for a journal of repute like Nature Communications, I must analyze the rigor of the study and seek responses to further improvement of the manuscript. Below are certain issues/points that need to be addressed.

Response:

Thanks for your positive comment and valuable suggestions.

1. The fabrication process relies on specific hygroscopic materials (e.g., diatomite-alginate shells), which may limit scalability or compatibility with other substances. Are these rails reusable? The six-cycle reusability is demonstrated. However, the transport dynamics are influenced by factors like humidity and temperature, which could impact performance in uncontrolled environments due to the impact of the ambient conditions on wettability and hysteresis.

Response:

We appreciate your thoughtful comment regarding the scalability, material compatibility, and environmental robustness of the hygroscopic microfiber rails.

Our study focuses on the development of shape-evolving microfiber rails (SEMRs) that achieve spontaneous and directional droplet transport via dynamically generated geometric gradients arising from droplet–surface interactions. While we employed a diatomite–alginate composite as a proof-of-concept hygroscopic material, the SEMR design is expected to be generalizable and compatible with a broad range of stimuli-responsive materials. For instance, solvent-responsive or thermoresponsive alternatives can be adopted to extend applicability beyond water droplets. Moreover, the microfluidic fabrication platform allows the use of diverse materials to tailor properties: hydrophilic silica nanoparticles for enhanced wettability, cellulose for improved mechanical strength, and magnetic particles for additional functionalities.

In response to your comment on reusability and environmental influence, we have expanded our characterization. As shown in Fig. 3d–e, SEMRs maintain consistent

deformation behavior over 100 hydration–dehydration cycles, and droplet transport performance remains stable after 30 days of ambient storage (Fig. 3f–g). Supplementary Fig. S16c–d further shows that higher humidity enhances droplet transport velocity by increasing surface wettability, while elevated temperature reduces velocity due to decreased surface hydration and increased contact angle hysteresis. These findings confirm that while environmental factors do influence performance, the SEMRs retain functional robustness under varying conditions.

Figure 3d-3g. **d**, SEMR undergoing hydration to evolve dynamic gradients and subsequent dehydration to restore its initial gradient-free topology. **e**, Deflection of microfibers during hydration and dehydration cycles. The deposited droplet volume is 0.4 μ L. **f**, Snapshots comparing the droplet transport performance of the SEMR after 30 days of storage. **g**, Plot of average transport velocity as a function of storage time, demonstrating long-term performance stability.

Supplementary Figure S16c-16d. Average transport velocity under varying humidity (c) and temperature (d).

We have updated the manuscript accordingly to include these expanded characterizations and discussions.

Lines 160–165, Page 7

“Upon dehydration, the SEMR passively returns to its original parallel, non-gradient configuration (Fig. 3d), dissipating the stored deformation energy and resetting the system for subsequent droplet transport. After 100 hydration–dehydration cycles, the deformation performance remains robust (Fig. 3e), indicating great reversibility and mechanical resilience. Moreover, transport efficiency remains stable after 30 days of storage (Fig. 3f and 3g).”

Lines 322–324, Page 13

“Environmental factors such as humidity and temperature also affect microfiber deformation and droplet transport dynamics (Supplementary Fig. S16), as they influence the wetting properties of the microfiber surface.”

Lines 440–448, Page 17

“Control of humidity

SEMRs were placed inside a semi-enclosed glass chamber, where the ambient humidity was precisely regulated using a combination of a humidifier and a dehumidifier. The environmental humidity was continuously monitored with a digital humidity sensor to ensure stable and accurate control.

Control of temperature

To regulate temperature, SEMRs were suspended above a hot plate, allowing uniform thermal exposure. The temperature was continuously monitored using a thermocouple positioned on SEMR to ensure precise thermal control throughout the experiments.”

2. The intricate design involving asymmetric microstructures requires precise control during manufacturing, which could increase production costs and complexity. For example, the $\beta=45^\circ$ in the illustrated case. Is it possible to obtain this level of precision?

Response:

Thank you for the insightful comment regarding the manufacturing precision of asymmetric microstructures. In our work, the microstructures are generated through a controlled microfluidic process in which the shape of microparticles embedded in the microfiber matrix can be finely tuned by manipulating droplet deformation and curing dynamics (**Supplementary Fig. S2**).

Specifically, during microfiber fabrication, oil microdroplets initially form spherical

shapes within the microchannel. Upon exiting, they deform asymmetrically due to shear-induced forces arising from differences in boundary conditions (e.g., flow rate and viscosity contrasts). These droplets gradually relax back toward spherical shapes over time due to surface tension. By selectively curing the droplets at specific stages of this shape evolution—i.e., varying the curing interval—we can precisely program the final microstructure. As shown in Supplementary Fig. S2b–c, increasing the curing time results in reduced asymmetry, with β_1 and β_2 converging toward $\sim 40^\circ$ after a 2-hour delay. This enables the fabrication of microstructures with well-controlled geometric asymmetry, e.g., the two inclined angles β_1 and β_2 .

Supplementary Figure S2: Tunable surface morphologies of microfibers. a, Tailoring microfiber morphology by adjusting microparticle shape through varying curing intervals. **b,** Comparison of microfiber morphologies before and after drying under different curing intervals. **c,** Tuning inclined angles (β_1 and β_2) achieved by controlling the curing interval.

We have added Supplementary Fig. S2 and updated the manuscript to include this explanation.

Lines 88–93, Page 3–4

“The constituent microfibers are fabricated using droplet microfluidic technology (Supplementary Fig. S1 and Supplementary Video 1), enabling precise control over their morphology by tuning the properties of the embedded microparticles (Supplementary Fig. S2). The eccentrically distributed microparticle arrays create a rugged side decorated with periodic micro-protrusions, while the encapsulating diatomite-alginate shell forms a flat spine on the opposite side (Fig. 2b).”

3. Focusing on Fig 1d) we see the rails, after the passage of the droplets, keep on deforming to a very large extent. Excessive spacing between microfiber rails during deformation can lead to droplet breakup due to Rayleigh-Plateau instability when the width-to-diameter aspect ratio exceeds a critical value.

Response:

Thank you very much for your insightful observation and valuable comment. You are right—excessive spacing between the microfiber rails during deformation can indeed trigger droplet breakup due to Rayleigh–Plateau instability, particularly when the width-to-diameter aspect ratio exceeds a critical threshold.

In response to this, we have clarified in the manuscript that the fate of the droplet—whether it undergoes directional transport or breaks up—is governed by the competition between the wicking speed of the droplet and the deformation speed of the microfiber rails. As you correctly pointed out, if the deformation outpaces the droplet’s motion, breakup is likely.

To elucidate this balance, we derived a phase map based on two characteristic timescales: one for wicking and one for deformation (Fig. 6f). This analysis, along with the expanded discussion in Lines 298–324 (Pages 12–13), shows that continuous droplet transport can be achieved by ensuring the wicking speed is faster than the deformation speed of the rails. Under this regime, even though the rail spacing continues to increase during deformation, the droplet moves away from the widest gap quickly enough to avoid breakup. As a result, the droplet maintains a nearly constant width during motion, as shown in Fig. 2d–e, indicating a well-coordinated interplay between rail deformation and droplet transport.

Figure 2d-2e. **d**, Schematic showing droplet-microfiber interactions during directional droplet transport. The droplet's dimensions are characterized by its width (W_{drop}) and length (L_{drop}). **e**, Plot showing the droplet's characteristic width (W_{drop}) and the SEMR's dynamic wedge angle (α) over time.

Lines 298–324, Pages 12–13

“The competition between wicking and microfiber deformation determines the droplet behavior. Insufficient wicking relative to deformation leads to excessive spacing between the microfiber rails, forcing the droplet to expand in width (W_{drop}). This increases the width-to-length aspect ratio, $C = W_{\text{drop}}/L_{\text{drop}}$, where L_{drop} is the droplet length (Fig. 2d). Once C exceeds a critical value, $C_{\text{criti}} \approx 2.41$ (Supplementary Fig. S13), the droplet spontaneously breaks apart due to Rayleigh-Plateau instability (Fig. 6e). Consequently, the droplet's fate on SEMR—directional transport or breakup—is dictated by the relative timescales of liquid wicking and microfiber deformation.

The characteristic wicking time (t_w) is defined as the time required for water to diffuse over a characteristic length along the microfiber (Supplementary Fig. S14). It scales as

$$t_w \sim \frac{L_c^2}{D}, \quad (3)$$

where L_c denotes the characteristic distance between two adjacent protrusions on the microfiber, and D is the water diffusion coefficient along the fiber (see Supplementary Fig. S3c, S3d). The characteristic deformation time (t_d) represents the interval from the onset of microfiber deformation to droplet breakup, approximated by

$$t_d \sim \frac{(\Omega C_{\text{criti}}^2)^{\frac{1}{3}} - s_0}{U_c} \quad (4)$$

where U_c is the microfiber separation speed (Supplementary Note S4). A linear correlation between t_w and t_d delineates the boundary between transport and breakup regimes in the phase diagram (Fig. 6f). Directional transport occurs when $t_d > t_w$,

indicating that wicking must outpace deformation to prevent droplet breakup. Enhancing the wicking process, such as by increasing the diffusion coefficient D , significantly improves transport speed, enabling continuous droplet motion (Fig. 6g) while preserving the characteristic step-wise behavior in transport velocity (Supplementary Fig. S15). Environmental factors such as humidity and temperature also affect microfiber deformation and droplet transport dynamics (Supplementary Fig. S16), as they influence the wetting properties of the microfiber surface.”

4. Further, the same figure shows a significant loss of droplet volume by wicking. In practical uses, the droplet would lose a lot of material in the arrangement. What would be the effective throughput as a function of ϵ .

Response:

Thank you very much for your valuable comment. Droplet volume loss on SERMs arises from two main factors: (1) water adsorption that drives microfiber swelling and deformation, and (2) the formation of a lubricating film along the droplet path.

In response to your comment, we experimentally quantified the liquid loss during transport (Supplementary Fig. S17). The results show that total liquid loss increases with transport distance. However, this effect can be reduced by optimizing the microfiber composition—for example, by changing the diatomite concentration.

Supplementary Figure S17: Liquid loss during droplet transport on SERMs. a, Two mechanisms of liquid loss: 1) water adsorption and 2) liquid film formation. **b,** Total liquid loss as a function of the transport distance. The microfiber length $L_f = 25$ mm. All droplets have an initial mass of 2.23 mg prior to transport. **c,** Tunable liquid loss per unit length (m/L_f) achieved by varying the diatomite concentration η .

We have included this analysis and discussion in the revised manuscript.

Lines 325–329, Page 13

“Although water adsorption and liquid film formation can result in minor liquid loss during droplet transport on SEMRs, this consumption can be minimized by tailoring the microfiber composition (Supplementary Fig. S17). Despite these challenges, SEMRs reliably support directional transport of microliter-scale droplets over extended distances (Supplementary Fig. S18 and Supplementary Video 5).”

5. Focusing on Fig. 2, it appears that the rails tend to shrink prior to the droplet reaching the point. Is it because of the strain in the wake of the droplet (behind the droplet)? The droplet creating upstream deformation seems to be non-causal.

Response:

Thank you very much for your insightful observation. The apparent rail shrinkage ahead of the droplet in Fig. 2 (Figs. 4 and 5 in the revised manuscript) is indeed very subtle, if present at all. This minor deformation is likely attributed to two factors: (1) capillary suction exerted by the approaching droplet, which can locally reduce the spacing between the two microfibers (as illustrated in Supplementary Fig. S5), and (2) liquid wicking that softens the microfiber material ahead of the droplet, leading to a slight reduction in stiffness and possible pre-deformation. However, we emphasize that this effect is not consistently observed across all trials. As shown in **Fig. 6a**, the dynamic evolution of the microfiber profile during droplet transport generally reveals no significant shrinkage ahead of the droplet.

Figure 6a. Dynamic evolution of the microfibers profile during droplet transport on SEMR.

6. Figure 3h is unclear. The deformation time is likely to be higher than the wicking time. That is not seemingly true for all the points.

Response:

Thank you for pointing out the ambiguity in the presentation of Fig. 3h (Fig. 6f in the revised manuscript). To clarify, the liquid wicking and microfiber deformation are two independent but competing processes that together govern droplet dynamics. We define the characteristic deformation time (t_d) as the duration from the onset of microfiber deformation to droplet breakup, and the characteristic wicking time (t_w) as the time required for water to diffuse along a characteristic length (L_c) on the microfiber surface (where L_c corresponds to the periodicity of the micro-protrusions). The ratio t_w/t_d thus captures the balance between liquid transport and structural actuation. Depending on experimental conditions (e.g., humidity, droplet volume, material properties), either process may dominate. This explains why t_d is not always greater than t_w for all data points.

Supplementary Figure S14: Definitions of characteristic times. The characteristic wicking time (t_w) corresponds to the time required for water to diffuse over a characteristic length (L_c) along the microfiber, where L_c refers to the distance between two adjacent protrusions on the microfiber. The characteristic deformation time (t_d) represents the interval from the onset of microfiber deformation to droplet breakup.

We have incorporated additional explanations and provided a schematic definition of t_w and t_d in **Supplementary Fig. S14**, and revised the corresponding discussion in the manuscript for clarity.

Lines 306–315, Pages 12–13

“The characteristic wicking time (t_w) is defined as the time required for water to diffuse over a characteristic length along the microfiber (Supplementary Fig. S14). It scales as

$$t_w \sim \frac{L_c^2}{D}, \quad (3)$$

where L_c denotes the characteristic distance between two adjacent protrusions on the microfiber, and D is the water diffusion coefficient along the fiber (see Supplementary Fig. S3c, S3d). The characteristic deformation time (t_d) represents the interval from the onset of microfiber deformation to droplet breakup, approximated by

$$t_d \sim \frac{(\Omega C_{\text{criti}})^{\frac{1}{3}} - s_0}{U_c} \quad (4)$$

where U_c is the microfiber separation speed (Supplementary Note S4).”

Once again, we sincerely thank you for your insightful comments and thoughtful consideration.

Response to the Comments by Reviewer #2

The present manuscript proposes an experimental investigation and proof-of-concept of a novel design for the directional transport of microfluidic droplets. In contrast with a permanent inhomogeneous geometry of the droplet's environment to drive its steady motion through surface energy minimization, the present process which relies on the use of flexible walls with complex macrostructures involving micro particle arrays and a hygroscopic shell that tends to absorb some of the droplet liquid and react to this absorption by asymmetric deformation. This wicking-induced deformation leads to confinement gradients or differential contact angle around the droplet, leading to the emergence of a driving force originating in this resulting asymmetric geometry.

The study is interesting and novel, the manuscript presents convincing details and validations of the concept. Beyond one major element that would absolutely need to be addressed by the authors before any decision regarding publication can be made, my comments are mostly minor and hopefully will help the authors enhance the clarity and pedagogy of their manuscript.

Response:

We highly appreciate your support and positive comments.

1. My most serious concern lies in some of the strong claims of the authors regarding the « self-adaptative » or « intelligent » nature of the material, or in the absence of steady gradients in their system. Admittedly, in its initial state, the microfluidic system only presents asymmetry at the microstructure level (if even) and reacts to the droplet presence by a deformation process that creates the necessary spatial gradient in an « unsteady » way. However, one should also argue that the gradient, despite invisible in the base state, remains nonetheless permanently encoded in the design of the system: repeating the experiment with different droplets seem to result in the same behavior, while one might have expected an « intelligent » system to « adapt » its reaction to the specific droplet or another form of non-unique stimulus. It would have been interesting for example to see if the system could be exploited for droplet sorting based on the droplet sizes, and its sensitivity. Also, the driving mechanism can still be associated with a « steady » gradient in free energy due to the irreversible transfer of liquid material in the wicking process.

Response:

Thank you very much for your thoughtful and constructive feedback. We fully acknowledge your concerns regarding the terminology used in our manuscript, particularly the use of “self-adaptive” and “intelligent.” Our original intention was to highlight two key features: (1) the *shape-evolving* behavior of the microfiber rails in response to droplet–surface interactions, and (2) the *tunable directional transport* enabled by pre-designed asymmetry in the system. However, we agree that these terms may overstate the capabilities of the system and could cause confusion.

Importantly, we would like to clarify the conceptual distinction between *gradient* and *asymmetry* in our system. The geometric gradient—i.e., the evolving spindle-shaped profile of the microfiber rails—is a transient feature induced by droplet wicking and fiber deformation. On its own, this symmetric gradient does not determine the transport direction. In contrast, the *pre-defined asymmetry* in the droplet deposit position or surface microstructures (i.e., $\beta_1 \neq \beta_2$) governs the directionality by biasing contact line dynamics during droplet motion. Thus, gradient and asymmetry are fundamentally different: the gradient arises from a responsive morphological change, while the asymmetry is encoded in the static design to provide direction selectivity.

In response, we have revised the manuscript to adopt more accurate terminology. Specifically, we have updated the title to “**Spontaneous droplet transport on shape-evolving microfiber rails**” and now refer consistently to “**SEMRs (shape-evolving microfiber rails)**” throughout the text. We have also clarified the system’s design principle and revised our discussion to avoid suggesting adaptive or intelligent behavior.

Lines 68–76, Page 3

“The design of SEMRs necessitates microfibers that exhibit (i) responsiveness to the transporting droplets (Fig. 1b), and (ii) structural asymmetry at the macro- or microscale (Fig. 1c). For instance, to facilitate the transport of water droplets, each microfiber consists of a hygroscopically responsive alginate–diatomite shell embedded with an array of eccentrically distributed microparticles (Fig. 1b). Upon contact with a water droplet, the hygroscopic swelling induces local deformation, giving rise to dynamically evolving geometric gradients. By introducing asymmetries—either in the droplet deposit position or in the surface microstructures (Fig. 1c)—these gradients can guide the droplet along programmable, steerable trajectories.”

Other detailed comments on the manuscript:

1. This is probably the result of « fitting » within the Nature Communications format, but I have had a very hard time reading the figures that include too many panels and use exceptionally small fonts making it difficult to decipher the information (and there is a lot on each figure). I would recommend a better hierarchization of the information, leading to (i) a careful selection of some panels that have a more minor role and could either be omitted or transmitted to the SI, (ii) the division of some of the most crowded figures (e.g. Figure 1) into distinct figures and (iii) more details on the figure panels once they have been increased in size — in particular the explaining schematics that play a critical role for the reader’s understanding.

Response:

Thank you for your valuable feedback. To improve clarity and readability, we have divided the original Figures 1 and 2 into separate, less crowded figures (now **Figures 1, 2, 4, and 5**), and enlarged all font sizes. We have also added explanatory schematics (e.g., **Fig. 1c** and **Supplementary Figs. S2a, S9a, S11a, and S14**) to enhance the reader’s understanding of key concepts.

2. Introduction: The authors do not precise what they mean by « trade-off between speed and distance » nor references or studies where such trade off was discussed or reported.

Response:

Thank you for your valuable comment. The “trade-off between speed and distance” refers to a limitation observed in droplet transport on static-gradient surfaces: high transport speeds typically require steep gradients, which limit the achievable distance, while long-distance transport demands shallower gradients, reducing the speed.

In line with your suggestion, we have now added explanatory context and cited relevant studies to support this point

Lines 37–41, Page 2

“However, these static-gradient approaches suffer from inherent limitations, including fixed transport directions and a trade-off between droplet velocity and transport distance. Specifically, high transport velocities require steep wetting gradients, while extended transport distances demand shallower gradients^{39,40}.”

References:

39 Malinowski, R., Parkin, I. P., & Volpe, G. *Advances towards programmable droplet transport on solid surfaces and its applications. Chem. Soc. Rev.* **49**,

7879-7892 (2020).

40 Ichimura, K., Oh, S. K., & Nakagawa, M. *Light-driven motion of liquids on a photoresponsive surface. Science* **288**, 1624-1626 (2000).

3. In some instances, in an effort to write the article in a catching and concise manner, some info needed to understand the meaning of the text is delayed to a later paragraph without any warning. For example, regarding the origin of the « asymmetry » mentioned around line 95: one only understands later on that the origin can take different forms: proximity to the attachment point of the fibers, microstructure asymmetry, etc...)

Response:

Thank you very much for your insightful comment. In response, we have added an explanatory schematic (**Fig.1c**) and expanded the discussion in the “Design principle” subsection (**Lines 68-76, Page 3**, as included in our response to your “most serious concern”) to clarify the pre-designed asymmetry.

Figure 1c. Asymmetry in either the droplet deposit position or surface microstructures, laying the basis for generating dynamically evolving gradients.

4. As discussed in my major comment, the origin of the asymmetry, despite the claim of the authors, is intrinsically encoded in the system’s design (through the asymmetric protrusions or the asymmetric distance to the bounding ends). How is that fundamentally-different from the classical « steady » gradient encoded in a microfluidic rail or anchor? The asymmetry just appears somewhat hidden temporarily and simply « revealed » but not « generated » by the droplet.

Response:

Thank you very much for your thoughtful comment. We agree that the asymmetry in our system is pre-encoded through design elements such as asymmetric microstructures or placement relative to fiber anchors. However, this asymmetry alone—similar to that on classical capillary ratchet surfaces—is insufficient to drive directional *droplet* transport without external input, as it does not inherently create a net driving force across the droplet body.

What distinguishes our system fundamentally is that the directional droplet motion emerges only when this encoded asymmetry interacts with the droplet through wicking-induced *deformation*, which dynamically creates a transient *geometric gradient*. Unlike surfaces with static gradients—where a steady spatial variation is fixed and always present—our system remains geometrically uniform (i.e., parallel rails) in its base state. The directional transport thus results not from a steady physical or chemical gradient, but from a time-dependent, droplet-triggered transformation of the system's geometry.

In response to your helpful suggestion, we have revised the manuscript to clearly explain the distinction between asymmetry and gradient and how both are leveraged in our SEMR design (**Lines 68-76, Page 3**). The revised content has been included in our response to your “most serious concern”.

5. A step critical to the robust functioning of the device is scarcely — if at all — described: dehydration. This is the step that will restore the system to its original state after the droplet has wicked it and gone by. One could interpret this as a « reloading » of the substrate's free energy so that it could drive a new droplet by monotonous energy transfer again. How is it achieved?

Response:

Thank you for your important question. The SEMR initially resides in a dehydrated, relaxed, and gradient-free state. Upon contact with a water droplet, the diatomite–alginate component absorbs water and swells asymmetrically due to the presence of non-hygroscopic, eccentrically distributed microparticles. This asymmetric swelling induces bending deformation of the microfiber, creating dynamic geometric gradients at the expense of droplet mass.

After the droplet completes its transport, the microfibers undergo dehydration through water evaporation, reversing the swelling-induced bending. This dehydration releases stored elastic energy, allowing the microfibers to recover their original straight configuration and restore the system to its initial state, effectively “reloading” the

substrate's free energy for subsequent droplet transport cycles.

Figure 3d-3e. d, SEMR undergoing hydration to evolve dynamic gradients and subsequent dehydration to restore its initial gradient-free topology. e, Deflection of microfibers during hydration and dehydration cycles. The droplet volume is 0.4 μ L.

To demonstrate this reversible process, we have added Fig. 3d–3e, and incorporated a discussion of the hydration–dehydration cycle in the revised manuscript

Lines 160–164, Page 7

“Upon dehydration, the SEMR passively returns to its original parallel, non-gradient configuration (Fig. 3d), resetting the system for subsequent droplet transport. After 100 hydration–dehydration cycles, the deformation performance remains robust (Fig. 3e), indicating great reversibility and mechanical resilience.”

6. Overall, I do not find the discussion of the timing aspect clear or convincing: on page 5, line 151, why does one side deflects « faster » (timing) than the other and not simply « more » (amplitude) ? What sets the timing?

Response:

Thank you very much for your valuable comment. When we used “faster,” we intended to convey that the deflection speed is higher on one side, which leads to a larger deflection amplitude and increased rail spacing. This interpretation aligns with your understanding.

To clarify, we have revised the relevant text accordingly.

Lines 201–203, Page 8

“For instance, when $\varepsilon > 0.5$, the microfiber on the droplet's left side deflects with a higher speed than the right, due to the reduced mechanical confinement imposed by the left-side fixed substrate (Fig. 4d).”

Lines 208–210, Pages 8–9

“When $\varepsilon > 0.5$, the speed ratio exceeds unity ($U_l/U_r > 1$), resulting in a larger rail spacing and a greater radius of curvature on the left ($R_l > R_r$).”

7. Line 182: the authors claim that ratchet geometries only facilitate liquid film spreading. What about Leidenfrost droplet propulsion on ratchet? Couldn't this be interpreted as direction droplet transport?

Response:

Thank you very much for your insightful comment. You are correct that Leidenfrost droplets can exhibit directional motion on ratchet surfaces. However, such transport relies on continuous external thermal input to sustain the vapor layer required for propulsion. In the absence of external actuation—such as heat, vibration, or electric fields—ratchet microstructures alone are insufficient to drive directional *droplet* transport due to the lack of a built-in gradient force.

In contrast, our study demonstrates spontaneous droplet transport on initially gradient-free surfaces without any external energy input. We have revised the relevant discussion to clarify this distinction.

Lines 41–42, Page 2

“In the absence of external controls, these surface gradients are indispensable, ...”

Lines 230–232, Pages 9–10

“While previous studies have utilized ratchet structures primarily to direct liquid film spreading^{1,23}, our approach leverages these asymmetric features to realize controlled directional transport of discrete droplets.”

8. page 7-8: I am confused by the discussion regarding the differential behavior of the two menisci on asymmetric protrusion. The current discussion suggests that both sides of the droplet act in the same direction regardless of the relative position of the two interfaces with respect to the protrusion pattern. I understand that the asymmetric microstructure may result for a single meniscus to create a horizontal force that is predominantly in one direction; yet I would anticipate that for some relative position in the protrusion and contact line, the direction of the forcing would be of opposite direction. Then, depending on the droplet size and relative position of the two sides, can there be a geometry-associated equilibrium?

Response:

Thank you very much for your insightful comment. Our force analysis focuses on

the *asymmetric receding behavior* of the contact line on the two sides of the droplet, which leads to *centroid displacement* rather than a geometry-associated equilibrium.

Specifically, when the ratchet angles are asymmetric ($\beta_1 < 90^\circ < \beta_2$), the obtuse-angle side (β_2) generates a downward-directed force component (F_{v-}) that promotes contact line retraction, while the acute-angle side (β_1) generates an upward-directed component (F_{v+}) that resists retraction. This imbalance in vertical force components causes faster receding on the β_2 side and a net shift of the droplet centroid (Fig. 5d).

To validate this mechanism, we fabricated structures with both β_1 and $\beta_2 < 90^\circ$, where both F_{v+} and F_{v-} point upward. In this case, retraction is suppressed on both sides, and the asymmetry no longer significantly affects droplet motion (**Supplementary Fig. S11**). This confirms that directional transport arises from asymmetric receding dynamics, not from a static force equilibrium.

Supplementary Figure S11: The effect of mound-like microstructure asymmetry on droplet transport performance. **a**, Schematics depicting the microscopic receding dynamics of droplets on SEMRs with both β_1 and β_2 less than 90° . The mound-like microstructures inhibit contact line retraction on both sides due to the upward-directed force components F_{v+} and F_{v-} . **b**, Variation in the transitional point position for droplet transport direction as a function of microstructure asymmetry ratio β_1/β_2 . The

microfiber length $L_f = 25$ mm. **c**, Transport distance versus time on SEMRs with varied β_1/β_2 . The volume of the water droplet is 2 μL .

We have revised the manuscript to clarify this mechanism and emphasize the role of force imbalance.

Lines 251–264, Page 10

“This asymmetric force distribution induces faster contact line motion over the left ratchet apex, while pinning occurs at the right apex, displacing the droplet's centroid asymmetrically to the right. A substantial difference between the two inclined angles, with $\beta_1 < 90^\circ < \beta_2$, is essential to trigger such asymmetric receding dynamics. In this case, downward-directed force component F_{v-} facilitates the descent of the contact line along the micro-protrusions on the obtuse-angle side (β_2), whereas the upward-directed force component F_{v+} on the acute-angle side (β_1) suppresses contact line descent. Consequently, ultra-fast receding of the contact line occurs along the β_2 side (Fig. 5d and Supplementary Video 4) immediately after crossing the left ratchet apex (position a–). This hypothesis is further validated by fabricating SEMRs with both β_1 and β_2 less than 90° , causing both F_{v-} and F_{v+} to point upward (Supplementary Fig. S11a), thereby inhibiting contact line retraction. Under these conditions, variations in the microstructure asymmetry (β_1/β_2) exhibit negligible impact on droplet transport dynamics, including directionality and velocity (Supplementary Fig. S11b and S11c).”

Once again, we sincerely thank you for your insightful comments and thoughtful consideration.

Response to the Comments by Reviewer #3

This manuscript proposes an innovative strategy of dynamically constructing structural gradients through asymmetric swelling of microfibers, theoretically overcoming the inherent trade-off between droplet transport velocity and distance in conventional intrinsic gradient systems. While the concept holds significant academic merit, several critical issues require substantial improvement:

Response:

Thanks for your positive comments and valuable suggestions.

1. Discrepancy Between Theoretical Promise and Experimental Performance. The reported droplet transport distance (centimeter-level) and velocity (millimeter/second scale) are notably inferior to state-of-the-art microfluidic systems. Dynamic contact line pinning induced by fiber surface roughness may be the primary limiting factor.

Response:

Thank you for your thoughtful comment. We agree that the current droplet transport distance and velocity are limited, primarily due to dynamic contact line pinning from fiber surface wettability. While our performance is indeed lower than state-of-the-art microfluidic systems, those systems typically rely on continuous external control (e.g., pressure, electric fields), which enable superior precision but at the cost of energy input and system complexity.

In contrast, our system achieves spontaneous, directional droplet transport on initially gradient-free surfaces without any external energy input. Compared to previous passive systems using wetting or curvature gradients, our approach offers comparable transport velocities (several millimeters per second) and longer transport distances (see **Supplementary Fig. S18b**).

Supplementary Figure S18b. Comparison of normalized transport distance (L/d) among different surfaces without external energy input. The parameter d denotes the droplet diameter, calculated as $d = (6\Omega/\pi)^{1/3}$, where Ω is the droplet volume.

We have added a sentence to the Conclusion (**Lines 390–392, Page 16**) to acknowledge this limitation and highlight directions for performance improvement.

“Further enhancement in transport performance can be achieved by engineering surface wettability to mitigate dynamic contact line pinning.”

2. Oversimplified Laplace Pressure Analysis. The current single-curvature (Rl) approach contravenes the fundamental Young-Laplace equation requiring orthogonal curvature components. Authors should: Develop mathematical models correlating Rl and Rr with microfiber deformation and contact angle (θ).

Response:

Thank you for raising this important point. We agree that the general Young–Laplace equation requires consideration of two orthogonal principal curvatures. In our revised analysis, we have incorporated both curvature components.

In our experiments, the droplet contact angle on the microfiber surface made from diatomite-alginate is zero (Supplementary Fig. S10). Thus, we consider a completely wetting droplet confined between two angled fibers, where the menisci on both sides of the droplet exhibit saddle-like curvature (**Supplementary Fig. S9a**). At the three-phase contact line on either side (points A and B), the Laplace pressure is given by:

$$P = P_0 + \gamma(1/R_h + 1/R_v),$$

where R_h is the in-plane (horizontal) radius of curvature ($-R_l$ or $-R_r$ for left and right sides, respectively), and $R_v = R_f$ is the out-of-plane (vertical) radius of curvature conforming to the fiber surface. The droplet fully wets the surface (contact angle $\theta = 0$), making the fiber-conforming curvature well approximated by the fiber radius R_f .

Thus, we have

$$P_a = P_A = P_0 + \gamma(1/R_f - 1/R_l),$$

$$P_b = P_B = P_0 + \gamma(1/R_f - 1/R_r).$$

Substituting these expressions, the pressure difference across the droplet becomes:

$$\Delta P = P_a - P_b = \gamma(1/R_r - 1/R_l).$$

This formulation incorporates both orthogonal curvature components and yields a

pressure difference solely governed by the difference in in-plane curvatures (R_l and R_r), since R_f is the same on both sides and cancels out.

We have experimentally validated this relationship using completely wetting liquid droplets spanning non-parallel fibers, where both fiber curvature and in-plane meniscus curvature are directly observed and measured. The results show strong agreement with the predicted pressure difference (**Supplementary Fig. S9b, S9c**).

Supplementary Figure S9: The Laplace pressure difference within the droplet during the transport process. a, Schematic of a completely wetting droplet confined between two angled microfibers. **b**, A plot of the transport velocity of ethanol droplets (3 μ L) between two non-parallel fibers (fiber radius $R_f = 0.35$ mm, fiber's wedge angle $\alpha = 4.7^\circ$) as a function of the Laplace pressure difference ΔP . **c**, Transport velocity of ethanol droplets (3 μ L) as a function of the fiber radius R_f under different ΔP .

In response to your comment, we have added **Supplementary Note S3 and Fig. S9** to include a more complete derivation highlighting the role of both curvature components, and revised the manuscript to clarify this point.

Lines 210–215, Page 9

“For a completely wetting droplet confined between two angled fibers, the Laplace pressure difference (ΔP) is independent of the fiber diameter and given by (Supplementary Fig. S9 and Note S3),

$$\Delta P = \gamma \left(\frac{1}{R_r} - \frac{1}{R_l} \right) \quad (2)$$

Given that $R_l > R_r$, the Laplace pressure difference is positive ($\Delta P > 0$), generating a net force from left to right and propelling the droplet in the positive direction.”

3. Neglect of Fluid Property Effects. Surface tension (γ) and viscosity (η) as key parameters remain unexplored.

Response:

Thank you very much for your insightful comment. In response, we have conducted additional experiments using sodium dodecyl sulfate (SDS) and carboxymethyl cellulose (CMC) solutions to systematically examine the effects of surface tension (γ) and viscosity (μ), respectively. As shown in the newly added **Fig. 3b and 3c** and **Supplementary Fig. S7**, we find that droplet transport velocity on SEMRs decreases with increasing γ and μ , confirming their influence on transport dynamics.

Figure 3b-3c. Effects of surface tension γ (b) and viscosity μ (c) on average transport velocity.

Supplementary Figure S7: Tuning of fluid property. a, Relationship between surface tension (γ) and concentration of sodium dodecyl sulfate (SDS). As the SDS

concentration increases from 0 to 0.2 wt%, γ decreases from $\sim 77 \text{ mN m}^{-1}$ to $\sim 32 \text{ mN m}^{-1}$. **b**, Relationship between dynamic viscosity (μ) and concentration of carboxymethyl cellulose (CMC). As the CMC concentration increases from 0 to 0.9 wt%, μ increases from $\sim 1 \text{ mPa s}$ to $\sim 365 \text{ mPa s}$.

Corresponding discussions have been added to the revised manuscript.

Lines 158–160, Page 7

“The SEMR further demonstrates the ability to transport droplets against gravity (Fig. 3a), and functions effectively across a broad range of droplet surface tensions γ (Fig. 3b; Supplementary Fig. S7a) and viscosities μ (Fig. 3c; Supplementary Fig. S7b).”

4. The discussion in Fig. 2e-f appears disconnected from the broader narrative of the manuscript. It is recommended that the authors repurpose this section to elucidate how the morphological features of SAMR in the Fig. 2a model (e.g., β_1 , the deflection speed ratio) govern liquid transport dynamics. A parametric study correlating these structural parameters with droplet acceleration (dv/dt) and Laplace pressure gradients would strengthen the mechanistic interpretation.

Response:

Thank you very much for your valuable comment. In response, we have reorganized the content by separating Fig. 2 into **Figs. 4 and 5** to better align the discussion with microfiber morphology. The original Fig. 2e–f is now presented as Fig. 5a–b.

We revised the discussion to explicitly relate the observed transport dynamics to key structural parameters—specifically, the asymmetric angles β_1 and β_2 . When $\beta_1 < 90^\circ < \beta_2$, the resulting force imbalance at the contact line (downward F_{v-} on β_2 and upward F_{v+} on β_1) drives asymmetric contact line motion and rightward centroid displacement, as illustrated in Fig. 5d and Supplementary Video 4.

To support this mechanism, we conducted control experiments using structures with both β_1 and $\beta_2 < 90^\circ$, where the force components are both upward, resulting in symmetric and suppressed retraction. These results are now shown in **Supplementary Fig. S11**, with corresponding discussion added (**Lines 251–264, Page 10**).

Supplementary Figure S11: The effect of mound-like microstructure asymmetry on droplet transport performance. **a**, Schematics depicting the microscopic receding dynamics of droplets on SEMRs with both β_1 and β_2 less than 90° . The mound-like microstructures inhibit contact line retraction on both sides due to the upward-directed force components F_{v+} and F_{v-} . **b**, Variation in the transitional point position for droplet transport direction as a function of microstructure asymmetry ratio β_1/β_2 . The microfiber length $L_f = 25$ mm. **c**, Transport distance versus time on SEMRs with varied β_1/β_2 . The volume of the water droplet is $2 \mu\text{L}$.

In addition, we examined the effect of microfiber wettability on droplet velocity. A higher surface wettability increases the diffusion coefficient (D) and enhances droplet velocity while preserving the step-wise motion behavior (**Supplementary Fig. S15**). We have also included a Laplace pressure difference analysis as in our response to Comment 2 (**Supplementary Fig. S9**).

Supplementary Figure S15: Transport velocity versus time on SEMR with varied diffusion coefficient D . The droplet volume is $2 \mu\text{L}$.

Corresponding discussions have been added to the revised manuscript.

Lines 251–264, Page 10

“This asymmetric force distribution induces faster contact line motion over the left ratchet apex, while pinning occurs at the right apex, displacing the droplet’s centroid asymmetrically to the right. A substantial difference between the two inclined angles, with $\beta_1 < 90^\circ < \beta_2$, is essential to trigger such asymmetric receding dynamics. In this case, downward-directed force component F_{v-} facilitates the descent of the contact line along the micro-protrusions on the obtuse-angle side (β_2), whereas the upward-directed force component F_{v+} on the acute-angle side (β_1) suppresses contact line descent. Consequently, ultra-fast receding of the contact line occurs along the β_2 side (Fig. 5d and Supplementary Video 4) immediately after crossing the left ratchet apex (position a–). This hypothesis is further validated by fabricating SEMRs with both β_1 and β_2 less than 90° , causing both F_{v-} and F_{v+} to point upward (Supplementary Fig. S11a), thereby inhibiting contact line retraction. Under these conditions, variations in the microstructure asymmetry (β_1/β_2) exhibit negligible impact on droplet transport dynamics, including directionality and velocity (Supplementary Fig. S11b and S11c).”

Lines 319–322, Page 13

“Enhancing the wicking process, such as by increasing the diffusion coefficient D , significantly improves transport speed, enabling continuous droplet motion (Fig. 6g) while preserving the characteristic step-wise behavior in transport velocity (Supplementary Fig. S15).”

5. Quantitative structure-performance relationships Insufficient Reliability Validation

Merely. 6 cycling tests inadequately demonstrate robustness. Required enhancements:
 I. 100-cycle positional drift statistics using optical positioning. II. Swelling reproducibility under varying humidity/temperature. III. 30-day storage performance degradation assessment.

Response:

Thank you for your valuable suggestions. In response, we have strengthened the reliability validation of SEMRs through the following experiments:

1. We conducted 100 hydration–dehydration cycles and quantified microfiber deformation using optical tracking (**Fig. 3d and 3e**). The results confirm consistent performance and suggest good reusability of SEMRs.
2. SEMRs retain high droplet transport efficacy after 30 days of ambient storage, with negligible degradation in performance (**Fig. 3f and 3g**).
3. We examined the influence of humidity and temperature on microfiber deformation and droplet transport performance (**Supplementary Fig. S16**). Low humidity level and higher temperature leads to lower surface hydration, thus less microfiber deformation and lower droplet transport velocity.

Figure 3d-3g. d, SEMR undergoing hydration to evolve dynamic gradients and subsequent dehydration to restore its initial gradient-free topology. e, Deflection of microfibers during hydration and dehydration cycles. The deposited droplet volume is 0.4 μ L. f, Snapshots comparing the droplet transport performance of the SEMR after 30 days of storage. g, Plot of average transport velocity as a function of storage time,

demonstrating long-term performance stability.

Supplementary Figure S16: Effects of the environmental conditions on microfiber deformation and droplet transport dynamics. a, b, Microfiber deformation under varying humidity levels (**a**) and temperatures (**b**). **c, d,** Average transport velocity under varying humidity levels (**c**) and temperatures (**d**).

Corresponding discussions have been added to the revised manuscript.

Lines 160–165, Page 7

“Upon dehydration, the SEMR passively returns to its original parallel, non-gradient configuration (Fig. 3d), resetting the system for subsequent droplet transport. After 100 hydration–dehydration cycles, the deformation performance remains robust (Fig. 3e), indicating great reversibility and mechanical resilience. Moreover, transport efficiency remains stable after 30 days of storage (Fig. 3f and 3g).”

Lines 322–324, Page 13

“Environmental factors such as humidity and temperature also affect microfiber deformation and droplet transport dynamics (Supplementary Fig. S16), as they influence the wetting properties of the microfiber surface.”

Lines 440–448, Page 17

“Control of humidity

SEMRs were placed inside a semi-enclosed glass chamber, where the ambient humidity was precisely regulated using a combination of a humidifier and a dehumidifier. The environmental humidity was continuously monitored with a digital humidity sensor to ensure stable and accurate control.

Control of temperature

To regulate temperature, SEMRs were suspended above a hot plate, allowing uniform thermal exposure. The temperature was continuously monitored using a thermocouple positioned on SEMR to ensure precise thermal control throughout the experiments.”

6. The liquid transport mechanism driven by swelling-induced deformation inherently introduces liquid loss during the transport process. The authors should address this issue in the discussion and implement necessary optimizations to the fibers, rather than intentionally avoiding it.

Response:

We appreciate your comment regarding the liquid loss. This loss arises from (1) water absorption driving microfiber deformation and (2) formation of a lubricating liquid film. We have quantified the mass loss during droplet transport (**Supplementary Fig. S17**), which increases with transport distance. Nevertheless, this effect can be significantly mitigated by tailoring microfiber composition—for example, by changing the diatomite concentration.

Supplementary Figure S17: Liquid loss during droplet transport on SEMRs. a, Two mechanisms of liquid loss: 1) water adsorption and 2) liquid film formation. **b**, Total liquid loss as a function of the transport distance. The microfiber length $L_f = 25$ mm. All droplets have an initial mass of 2.23 mg prior to transport. **c**, Tunable liquid loss per unit length (m/L_f) achieved by varying the diatomite concentration η .

Importantly, despite partial consumption of liquids, SEMRs enable robust,

directional transport of microliter-scale droplets over extended distances (Supplementary Fig. S18, Supplementary Video 5). Moreover, this liquid-consumption-driven mechanism offers functional benefits in applications such as particle coating, oil-phase enrichment, and localized reagent delivery, as demonstrated in Fig. 7f, 7h, and Supplementary Fig. S19 (to be displayed in our next response to Comment 7).

We have added these data and contextual discussions to the revised manuscript.

Lines 325–329, Page 13

“Although water adsorption and liquid film formation can result in minor liquid loss during droplet transport on SEMRs, this consumption can be minimized by tailoring the microfiber composition (Supplementary Fig. S17). Despite these challenges, SEMRs reliably support directional transport of microliter-scale droplets over extended distances (Supplementary Fig. S18 and Supplementary Video 5).”

7. The authors propose several applications based on this model in the manuscript, but these applications lack innovation, and their practicality is questionable.

Response:

Thank you for your thoughtful comment. The core innovation of this work lies in enabling spontaneous, directional droplet transport on initially gradient-free surfaces, driven by droplet–microfiber interaction and dynamically evolving geometry. This mechanism overcomes the limitations of static-gradient systems, allowing for tunable directionality and decoupled control of transport speed and distance.

The proposed applications—such as analytical chemistry, cargo transport, electronic circuits, and medical diagnostics—operate without external actuation, distinguishing them from prior approaches. For example, unlike previous fluidic switches requiring continuous liquid supply for film spreading, our SEMRs achieve the same function via discrete droplet transport, significantly reducing fluid consumption (Fig. 7g).

To further demonstrate utility, we explored two additional applications: (1) oil-phase enrichment through controlled water extraction from oil-in-water emulsions, and (2) uniform particle deposition enabled by residual film left during droplet transport (Supplementary Fig. S19). These examples illustrate the broader potential of SEMRs in material processing and microfluidic control.

Supplementary Figure S19: Oil-phase enrichment and microparticle coating enabled by SEMRs. **a**, Schematic and experimental demonstration of SEMR-mediated oil enrichment through controlled water extraction from oil-in-water emulsions during droplet transport. **b**, Uniform microparticle deposition facilitated by the residual liquid film. The fluorescence photograph (bottom) confirms homogeneous microparticle distribution on SEMR surfaces after droplet transport.

We have incorporated these new results and supporting discussions in the revised manuscript.

Lines 348–352, Page 14

“This multiphase manipulation capability is preserved even at smaller scales. For instance, SEMRs facilitate oil-phase enrichment via controlled water extraction from oil-in-water emulsions and enable uniform deposition of microparticles on microfiber surfaces following the transport of a suspension droplet (Supplementary Fig. S19).”

Once again, we sincerely thank you for your insightful comments and thoughtful consideration.

Response to the Comments by Reviewer #1

I am very satisfied with the modifications and believe that the manuscript is fit for publication now. I would again like to emphasize on the elegance and simplicity of the methodology and believe the manuscript will be a worth archival material for Nat Comm.

Response:

Thanks for your positive feedback and for recommending our manuscript for publication in *Nature Communications*.

Response to the Comments by Reviewer #2

I thank the authors for taking the time and care to carefully address and respond to my comments on their original submission. I believe these changes strongly enhance the manuscript's clarity and impact and am therefore happy to recommend the publication of this manuscript.

Response:

Thanks for your positive feedback and for recommending our manuscript for publication in *Nature Communications*.

Response to the Comments by Reviewer #3

While the revised manuscript represents a significant improvement over the previous version, with notably enhanced figures and text, as well as substantial additional data supporting the claims, I regret that I cannot recommend publication in Nature Communications at this time. My primary concern relates to the core innovation claimed – namely, overcoming the inherent trade-off between droplet transport velocity and distance in conventional intrinsic gradient systems without external energy input – which motivated my previous recommendation for major revision.

However, the optimized performance metrics presented in this revision, specifically the achieved centimeter-level transport distance and millimeter-per-second velocity, do not convincingly demonstrate the claimed breakthrough in decoupling distance and velocity. This assessment is based on several key points:

Response:

Thank you very much for your thoughtful and constructive feedback. We fully acknowledge the concern regarding our original claim of “overcoming the inherent trade-off between droplet transport velocity and distance”. Our intention in the previous manuscript was to highlight that our system enables **independent tuning of transport velocity and distance**, a level of control not readily achievable in conventional static-gradient surfaces. We agree that our earlier wording may have overstated the extent of decoupling and could have caused confusion. Accordingly, we have revised the discussion and associated text to clarify that our contribution is **not an absolute transcendence of all physical limits**, but rather a demonstration of **on-demand, independent tuning of these parameters**.

More importantly, the novelty of our work extends well beyond the velocity–distance relationship. Conventional static-gradient surfaces are inherently limited in controlling the **direction of droplet transport** on initially non-gradient surfaces, which is fundamentally difficult, if not impossible, without external energy input. Our **shape-evolving microfiber rails (SEMRS)** overcome this limitation by dynamically generating gradients in response to droplet interactions, enabling **selective and independent control of droplet direction, velocity, distance, and volume** simultaneously. This capability represents a conceptual advance that goes far beyond the coupled velocity-distance issue emphasized in prior studies.

To further strengthen the manuscript, we have revised the Introduction to clearly highlight the **key contributions and conceptual novelty** of our study. In addition, we have incorporated **surface modifications of SEMRs** to demonstrate enhanced transport performance, showing both increased transport distance and velocity (see the point-by-point response below). These results underscore the **high compatibility and versatility** of SEMRs, and further illustrate their potential as a general platform for intelligent liquid transport.

Lines 46–68, Pages 2-3

“Despite significant progress, static-gradient surfaces remain fundamentally constrained. Because their gradients are permanently fixed once fabricated, they can typically improve only one or a limited subset of performance metrics—such as directionality, velocity, or distance—often at the expense of others⁴²⁻⁴⁵. Consequently, one must compromise between competing design goals, as direction, speed, distance, and volume of droplets cannot be independently or dynamically controlled. Overcoming these constraints requires strategies that transcend the inherent limitations of static-gradient designs. Yet, a key question remains unresolved: can directional droplet transport be realized on surfaces devoid of intrinsic gradients?”

We introduce shape-evolving microfiber rails (SEMRs), an initially non-gradient system designed to enable spontaneous droplet transport. Unlike conventional static-gradient surfaces, SEMRs self-generate and continuously evolve geometric gradients through droplet-surface interactions, establishing a feedback-driven mechanism for motion. This self-adaptive behavior allows for on-demand tuning of all key transport parameters—direction, velocity, distance, and volume of droplets—within a single system. By integrating multiple performance controls into a unified and reconfigurable surface, SEMRs overcome the inherent limitations of static designs. With their dynamic adaptability and broad functionality, SEMRs offer a versatile foundation for next-generation microreactors, fluidic logic circuits, diagnostics, and autonomous liquid handling systems. This study thus establishes a paradigm of self-evolving liquid transport surfaces, bridging the gap between passive gradient engineering and active fluidic control.”

1. The supplementary video (S18) appears to contradict the stated transport distance, as measurement via the provided scale bar indicates droplet movement of approximately 7 mm over 17.4 seconds rather than the reported 12 mm, leading to transport speed

overestimation.

Response:

Thank you for pointing out this important inconsistency. We believe you were referring to Supplementary Fig. 18, as Supplementary Video 18 does not contain a scale bar. The discrepancy in Supplementary Fig. 18 arose from an incorrect scale bar, which was inadvertently altered during figure formatting. We apologize for this oversight.

To correct this, we have updated **Supplementary Fig. S18** to accurately reflect the true dimensions. In the revised figure, the droplet is shown advancing ~ 12 mm over 17.4 s, consistent with the reported transport distance and speed. To facilitate direct verification, we have included the microfiber diameter and pipette tip diameter as a scale reference (**Fig. R1**).

Additionally, to prevent similar errors, we have systematically reviewed all figures and data in the manuscript, verifying the accuracy of every scale bar and measurement. The original raw data have been retained and are available for review upon request.

Supplementary Figure S18: Optical images of a 2 μ L water droplet transporting from the deposit position to the right end. The droplet advances ~ 12 mm in 17.4 s.

Figure R1. Optical images of (a) the microfiber (diameter of 520 μ m) and (b) the

pipette (10 μ L, tip diameter of 1 mm) served as scale references.

2. Furthermore, comparison with prior art is insufficient and lacks rigor; crucially, multiple studies achieving droplet transport without external energy input report significantly superior performance, such as *J. Mater. Chem. A*, 2023, 11, 10164–10173 and *ACS Appl. Mater. Interfaces* 2017, 9, 9213–9220 (exceeding results by 1-2 orders of magnitude), and *Adv. Funct. Mater.* 2023, 33, 2212217 (analogous dual-fiber system showing several-fold higher metrics).

Response:

Thank you for highlighting these highly relevant studies. We agree that a more rigorous comparison is essential, and we have now added a detailed performance comparison in **Fig. 7d (Supplementary Fig. S19)**, including the cited references and other representative works. While the prior studies achieve remarkable droplet velocity and distance using static surface designs, our **SEMR system introduces a fundamentally dynamic mechanism**, enabling flexible, on-demand control over droplet transport direction, velocity, distance, and volume.

Importantly, **surface modifications further enhance SEMR performance**. Coating microfibers with non-hygroscopic materials minimizes water loss, allowing reliable transport of **nanoliter droplets** over long distances (**Fig. 7c**). By contrast, many conventional static-gradient systems rely on self-lubrication (*J. Mater. Chem. A*, 2023, 11, 10164–10173), large enough driving forces (*ACS Appl. Mater. Interfaces* 2017, 9, 9213–9220), or geometric constraints (*Adv. Funct. Mater.* 2023, 33, 2212217), which limit their performance and require larger droplets. The SEMR's **initially gradient-free, deformable microfibers** can dynamically generate geometric gradients in response to droplet interactions, enabling high velocity (up to ~ 15 mm/s, **Fig. 7a,b**) and long transport distance while maintaining compatibility with a broad range of droplet volumes (nL– μ L, **Fig. 7c,d**).

These capabilities allow SEMRs to achieve the **leading transport distance per droplet volume** among reported systems (**Fig. 7d**), highlighting a unique combination of versatility and efficiency that is unattainable in previous static designs.

Figure 7: The versatility of SEMRs. **a**, Time-sequence images showing the continuous transport of multiple drops (2 μL) along the SEMR. **b**, Variations in droplet transport velocity and water loss versus the number of droplets transported. **c**, Optical images showing the transport of a 0.5 μL water droplet along the coated SEMR. In the upper image, the dashed line marks the boundary between regions with (right) and without (left) the non-hygroscopic coating. **d**, Comparison of transport distance per droplet volume (L/Ω) among surfaces without external energy input. The green, red, and yellow columns indicate the reported transport performances for surfaces with wettability gradients, geometry structures, and their combinations, respectively (Supplementary Fig. S19).

We have incorporated these new results and discussions in the revised manuscript.

Lines 342–359, Pages 13-14

“Although water adsorption and liquid film formation can induce liquid loss during transport (Supplementary Fig. S17), lubricated SEMRs markedly enhance droplet transport performance. Prewetting has previously been employed to boost droplet velocity on dry surfaces^{36,48}, and a similar effect is observed here. Following the passage of a single droplet, the initially dry SEMR becomes lubricated and deformed, enabling the continuous transport of subsequent droplets at velocities up to $\sim 15 \text{ mm s}^{-1}$ with water loss reduced to $\sim 10\%$ (Fig. 7a,b). Furthermore, the liquid loss can be

significantly minimized—to approximately $0.01 \mu\text{L mm}^{-1}$ (volume per unit transport distance)—by coating part of the SEMR with a non-hygroscopic material (Fig. 7c). This modification simultaneously preserves the hygroscopically induced deformation required for gradient evolution while providing a non-hygroscopic property for long-distance transport of minute droplets. Compared with the uncoated SEMR, which transports microliter-scale droplets (Supplementary Fig. S18 and Supplementary Video 5), the coated SEMR enables reliable nanoliter droplet transport (Fig. 7c) and achieves a substantially higher transport distance per droplet volume (Fig. 7d and Supplementary Fig. S19). These results highlight the high material compatibility and design flexibility of SEMRs, demonstrating that the transport performance can be finely tuned through simple surface modifications.”

Lines 378–379, Page 15

“Notably, a peak transport velocity of $\sim 21 \text{ mm s}^{-1}$ is attained during this droplet-chasing process.”

Lines 430–434, Page 17

“This dynamically-evolving mechanism allows for on-demand and independent tuning of transport direction, velocity, and distance, thereby overcoming the inherent limitations of conventional static-gradient surfaces. Moreover, the transport performance can be further enhanced through surface modifications that tailor droplet–surface interactions.”

Lines 480–483, Page 19

“Non-hygroscopic coating of SEMRs

The microfluidically-generated microfibers were uniformly coated with a photocurable resin, followed by UV curing for 30 s. Subsequently, they were modified with TM-MES and left undisturbed for 24 h.”

3. Additionally, the mechanism's critical reliance on droplet-induced fiber swelling introduces unaddressed limitations: small droplets may deplete due to absorption/swelling during long-distance transport; fiber recovery via evaporation imposes fundamental constraints on continuous/batch transport; and droplet size sensitivity may preclude practical applicability. These newly identified limitations concerning efficiency, operational continuity, and droplet size compatibility represent significant drawbacks directly impacting the claimed novelty and utility of the SEMR approach.

Response:

We appreciate your insightful comment regarding limitations associated with droplet-induced swelling and transport efficiency. The SEMR's inherent versatility and compatibility provide multiple strategies to overcome these challenges.

Droplet loss has been effectively minimized by coating the microfibers with a non-hygroscopic material, reducing liquid loss to $\sim 0.01 \mu\text{L mm}^{-1}$ (volume per unit transport distance) and enabling reliable long-distance transport of **nanoliter droplets (Fig. 7c)**, a size range substantially smaller than those handled by many previous systems that are limited to microliter volumes. **Operational continuity** is maintained through the transport of multiple droplets (**Figs. 7a,b and 8**), overcoming constraints imposed by fiber recovery. Recovery efficiency can be further enhanced by moderately heating the microfibers, accelerating dehydration and shortening reset times (**Fig. 3e**).

The SEMR's initially gradient-free design allows compatibility across a **broad droplet-volume range** (nL– μL), demonstrating tunable, versatile transport without compromising performance. The wide array of applications illustrated in **Fig. 8** further demonstrates that the approach does not compromise functionality or applicability. Collectively, these results highlight that the novelty of SEMRs lies in their **dynamic tunability, operational versatility, and broad compatibility**, which surpass the capabilities of previous static-gradient systems.

Figure 3e. Effects of ambient temperature on dehydration duration of SEMR.

In addition to the specific responses provided for Comment 2, Fig. 3e and a related discussion have been incorporated into the revised manuscript.

“The rate of this dehydration-driven reset is highly dependent on ambient temperature, with higher temperatures leading to faster recovery (Fig. 3e).”

4. Therefore, based on unresolved concerns regarding performance benchmarks, data accuracy, and intrinsic system limitations, I conclude the manuscript in its current form does not meet Nature Communications' high-impact threshold.

Response:

We sincerely thank you for the rigorous assessment and the insightful feedback, which have been invaluable in strengthening our manuscript. We understand the concern regarding the need to clearly distinguish the conceptual innovation of our work from performance metrics.

The core contribution of our study lies in demonstrating a **dynamically geometry-evolving platform with initially gradient-free surfaces for spontaneous droplet transport**. While conventional systems achieve high static performance for specific metrics, our core contribution lies in enabling **spontaneous droplet motion with independent and tunable control over multiple parameters**, including direction, velocity, transport distance, and droplet volume. In this revision, we have:

1. Corrected all data and scale inconsistencies to ensure accuracy.
2. Conducted rigorous comparisons with relevant prior studies, highlighting the advantages of SEMRs over static-gradient systems.
3. Demonstrated enhanced performance through material and structural modifications, including non-hygroscopic coatings to reduce liquid loss and restricted microfiber deformation for continuous transport.
4. Validated broad operational versatility, including transport of nanoliter-to-microliter droplets and maintenance of functionality across multiple sequential droplet operations (Figs. 7–8).

These revisions clarify that the novelty of SEMRs lies not only in performance but also in **dynamic tunability, programmable directionality, operational continuity, and broad applicability**, establishing a paradigm for intelligent droplet transport.

Once again, we sincerely thank you for your insightful comments and thoughtful consideration.